# Large Language Models for Scientific Idea Generation: A Creativity-Centered Survey

**Fatemeh Shahhosseini**                    *f.shahhosseini.20@gmail.com*
*Department of Computer Engineering*
*Sharif University of Technology*

**Arash Marioriyad**                         *arashmarioriyad@gmail.com*
*Department of Computer Engineering*
*Sharif University of Technology*

**Ali Momen**                                *ali.momen8998@gmail.com*
*Department of Computer Engineering*
*Iran University of Science and Technology*

**Mahdieh Soleymani Baghshah**               *soleymani@sharif.edu*
*Department of Computer Engineering*
*Sharif University of Technology*

**Mohammad Hossein Rohban**[*]                *rohban@sharif.edu*
*Department of Computer Engineering*
*Sharif University of Technology*

**Shaghayegh Haghjooy Javanmard**[*]          *sh__haghjoo@med.mui.ac.ir*
*Department of Physiology*
*Isfahan University of Medical Sciences*

**Reviewed on OpenReview:** *https://openreview.net/forum?id=9lWojZKMjt*

## Abstract

Scientific idea generation is central to discovery, requiring the joint satisfaction of novelty and scientific soundness. Unlike standard reasoning or general creative generation, scientific ideation is inherently open-ended and multi-objective, making its automation particularly challenging. Recent advances in large language models (LLMs) have enabled the generation of coherent and plausible scientific ideas, yet the nature and limits of their creative capabilities remain poorly understood. This survey provides a structured synthesis of methods for LLM-driven scientific ideation, focusing on how different approaches trade off novelty and scientific validity. We organize existing methods into five complementary families: External knowledge augmentation, Prompt-based distributional steering, Inference-time scaling, Multi-agent collaboration, and Parameter-level adaptation. To interpret their contributions, we adopt two complementary creativity frameworks: Boden's

---

[*]Corresponding Author

taxonomy to characterize the expected level of creative novelty, and Rhodes' 4Ps framework to analyze the aspects or sources of creativity emphasized by each method. By aligning methodological developments with cognitive creativity frameworks, this survey clarifies the evaluation landscape and identifies key challenges and directions for reliable and systematic LLM-based scientific discovery.

## 1 Introduction

Scientific discovery has long stood at the frontier of human progress, from uncovering the laws of physics to developing transformative medicines. At the heart of this process lies scientific idea generation—the ability to propose hypotheses that are not only new but also scientifically meaningful. In cognitive science, this ability is captured by *creativity*, which defines as requiring two jointly necessary components: *novelty* and *valueness*(Boden, 2004). Novelty refers to the degree to which an idea departs from existing knowledge or established patterns, while valueness concerns whether the idea is correct, plausible, feasible, or useful within a scientific context. Novelty without value results in arbitrary or incoherent ideas; value without novelty yields incremental or redundant contributions. Scientific ideation, therefore, emerges only when these two criteria coexist.

Recent advances in large language models (LLMs) offer an unprecedented opportunity to augment this process. On the *valueness* side, research in *reasoning and factuality* has improved model reliability. Techniques such as Chain-of-Thought prompting (Wei et al., 2022), Self-Consistency decoding (Wang et al., 2023c), Tree-of-Thoughts (Yao et al., 2023), and iterative refinement methods like Self-Refine (Madaan et al., 2023) enhance structured reasoning. Inference-time scaling approaches, from repeated sampling (Brown et al., 2024) to advanced search (Sprueill et al., 2023) and frontier RL post-training models (DeepSeek-AI et al., 2025; Yin et al., 2025b), further demonstrate that compute can unlock deeper reasoning abilities. Complementary strategies including retrieval-augmented generation (RAG) (Lewis et al., 2020) and agentic tool use integrate external knowledge sources and fact-checking pipelines to ensure factual grounding (Schick et al., 2023). Collectively, these advances provide a foundation for promoting valueness and correctness in scientific ideation.

In contrast, efforts explicitly targeting *novelty* remain more limited and domain-dependent. Alignment-oriented methods push LLMs toward greater originality. Extensions of direct preference optimization (DPO) such as Creative Preference Optimization (CRPO) (Ismayilzada et al., 2025a) and Diverse Preference Optimization (DivPO) (Lanchantin et al., 2025) explicitly inject signals for novelty and diversity into the optimization process. However, such methods often excel in domains like storytelling or open-ended writing, where factual complexity is limited, but struggle to generalize to science, where creativity must remain tightly coupled with empirical soundness.

The convergence of reasoning and generative creativity highlights why scientific idea generation is uniquely challenging: it is inherently a multi-objective problem where calls for a deeper understanding of creativity itself. Creativity, a higher-order cognitive function of the human mind that LLMs do not yet possess (Liu et al., 2025a), remains poorly understood within this emerging field.

To ground our analysis, we turn to cognitive science, which offers robust frameworks for studying and categorizing creativity and helps clarify which cognitive dimensions are captured by current LLM-based approaches and which remain underexplored. Rhodes' seminal *4Ps* framework (Rhodes, 1961) conceptualizes creativity as the interaction among four dimensions: the *person* (the individual

or system generating ideas), the *process* (the cognitive or algorithmic mechanisms involved), the *press* (the surrounding environment and context), and the *product* (the evaluable outcome). While all four are jointly constitutive of creativity, later work emphasizes that the *product* is typically the dependent, measurable outcome, whereas *person*, *process*, and *press* act as the primary sources shaping its emergence (Kozbelt et al., 2010; Gruszka & Tang, 2017).

Complementing this perspective, Boden (Boden, 2004) distinguishes three levels of creative output: *combinatorial* creativity, arising from novel recombinations of existing concepts; *exploratory* creativity, which results from structured search within a given conceptual space; and *transformational* creativity, which alters or expands the space itself. In this survey, we use Boden's taxonomy to characterize the expected creative potential of LLM-based scientific ideation methods according to their dominant mechanisms. Methods centered on recombination are thus expected to yield combinatorial creativity, those enabling systematic exploration to favor exploratory creativity, and those that challenge established assumptions to potentially approach transformational creativity. These assignments are conceptual ,reflecting tendencies rather than guaranteed outcomes.

Building on these foundations, this survey—unlike existing works that primarily emphasize engineering pipelines, agent architectures, or task coverage (Zheng et al., 2025; Luo et al., 2025; Eger et al., 2025; Han & Cheng, 2025; EmergentMind Research Group, 2024; Ren et al., 2025)—examines how approaches in LLM-driven scientific discovery manage the dual objectives of scientific valueness and novelty. We propose that these objectives can be systematically understood through the lens of *output creativity levels* (combinatorial, exploratory, transformational) and *creativity sources* (person, process, press), as illustrated in Figure 11, which maps current methods to their primary sources of creativity. Although Boden's and Rhodes' frameworks are classical, their intentionally high-level and orthogonal dimensions allow them to map naturally onto modern LLM-based systems. The 4Ps emerged empirically from analyses of creativity definitions and were independently rediscovered across multiple studies, supporting their robustness as an organizing framework (Jordanous, 2012). Unlike alternative creativity-level frameworks such as the 4C model (Kaufman & Beghetto, 2009), whose categories span everyday and personal creativity (e.g., mini-c and little-c), Boden's taxonomy is tailored to research-level idea generation, where novelty arises through recombination, structured exploration, or transformation of conceptual spaces; as a result, these frameworks continue to serve as foundational baselines in recent studies of LLM creativity (Nagarajan et al., 2025; Gu et al., 2025; Jackson et al., 2025; Ismayilzada et al., 2025b).

To make this analysis concrete, as illustrated in Figure 3, we categorize existing methods into five complementary families according to their primary mechanisms for influencing novelty and valueness in LLM-based ideation:

1. **Knowledge and retrieval augmentation:** A central line of work enhances LLMs with relational domain knowledge, compensating for the limitations of static pre-training. Adding related literature or factual resources into the model's context not only grounds outputs in established knowledge and reduces hallucinations, but also acts as a source of creativity infusion from the environment. In 4Ps, this reflects the *press*, balancing correctness with novelty derived from recombining known elements. Such methods are most likely to foster combinatorial creativity. We expand on these approaches in Section 2.

2. **Prompt-based distributional steering:** Another family of approaches steers LLMs toward more original ideas through input manipulations. Manipulations like role priming,

constraint-based prompting, or structured scaffolds encourage less typical outputs without parameter changes. Rooted mostly in the *press*, they often yield combinatorial creativity but can also broaden into exploratory forms when prompts guide a systematic exploration. Details and more examples appear in Section 3.

3. **Search and sampling expansions:** Inference-time scaling—via iterative refinement or branching exploration — expand the capabilities of LLMs beyond single-pass decoding by enabling dynamic exploration and without additional model training. By systematically enlarging the idea search space, these methods increase the chance of exploratory creativity, though correctness depends on strong evaluation signals. This family maps most closely to the *process* dimension. We discuss them further in Section 4.

4. **Multi-agent and deliberative systems:** Beyond single-agent reasoning, multi-agent systems emulate the collaborative dynamics of scientific teams. Beyond automation, such setups—featuring debate, critique, or role specialization—foster emergent interactions that extend reasoning beyond individual limits, often surfacing cross-disciplinary boundaries or unconventional ideas. These dynamics promote critical thinking, analogy-making, and out-of-the-box problem solving—hallmarks of transformational creativity. This line of work highlights the process dimension and is discussed further in Section 5.

5. **Parameter adaptation and learning:** Fine-tuning, reinforcement learning, and hybrid alignment approaches directly modify model's parameters, internalizing new strategies and knowledge. Acting on the *person* dimension, as models internalize reasoning patterns, the likelihood of reaching closer to higher-level creative outcomes may increase. We return to these methods in Section 6.

This taxonomy is derived by synthesizing recurring design patterns in the LLM-based scientific discovery literature and is intended to be general and extensible rather than exhaustive; hybrid and future methods may span or extend multiple categories. Table 3 summarizes how the five method categories are empirically observed or theoretically expected to influence the creativity.

Beyond methodological categorization, this survey also examines how *evaluation* practices shape and constrain scientific creativity. In particular, we analyze the current landscape of metrics and methods for assessing generated ideas, highlighting persistent challenges of subjectivity, reliability, and comparability. These issues map naturally to the *product* dimension of Rhodes' 4Ps framework, reflecting how scientific ideas are judged once produced. We discuss evaluation metrics in detail in Section 7.

Finally, we outline open directions in Section 9. Current methods tend to remain at the combinatorial or exploratory level, with transformational creativity still elusive. Much progress has been made from the "process" and "press" perspectives, yet the "person" and "product" dimensions remain underexplored. For example, shifting from idea-level search to agent-level search, as it is done in recent open-ended methods (Zhang et al., 2025a), may unlock richer person-level creativity. At a deeper level, the auto-regressive training paradigm or llm attention-based architecture itself may impose structural limits on the creative potential of LLMs in science (Nagarajan et al., 2025). On the product side, the lack of standardized metrics and benchmarks for evaluating generated scientific ideas leaves creativity assessment vague and subjective. Addressing these gaps could move us closer to realizing LLMs as true partners in scientific discovery.

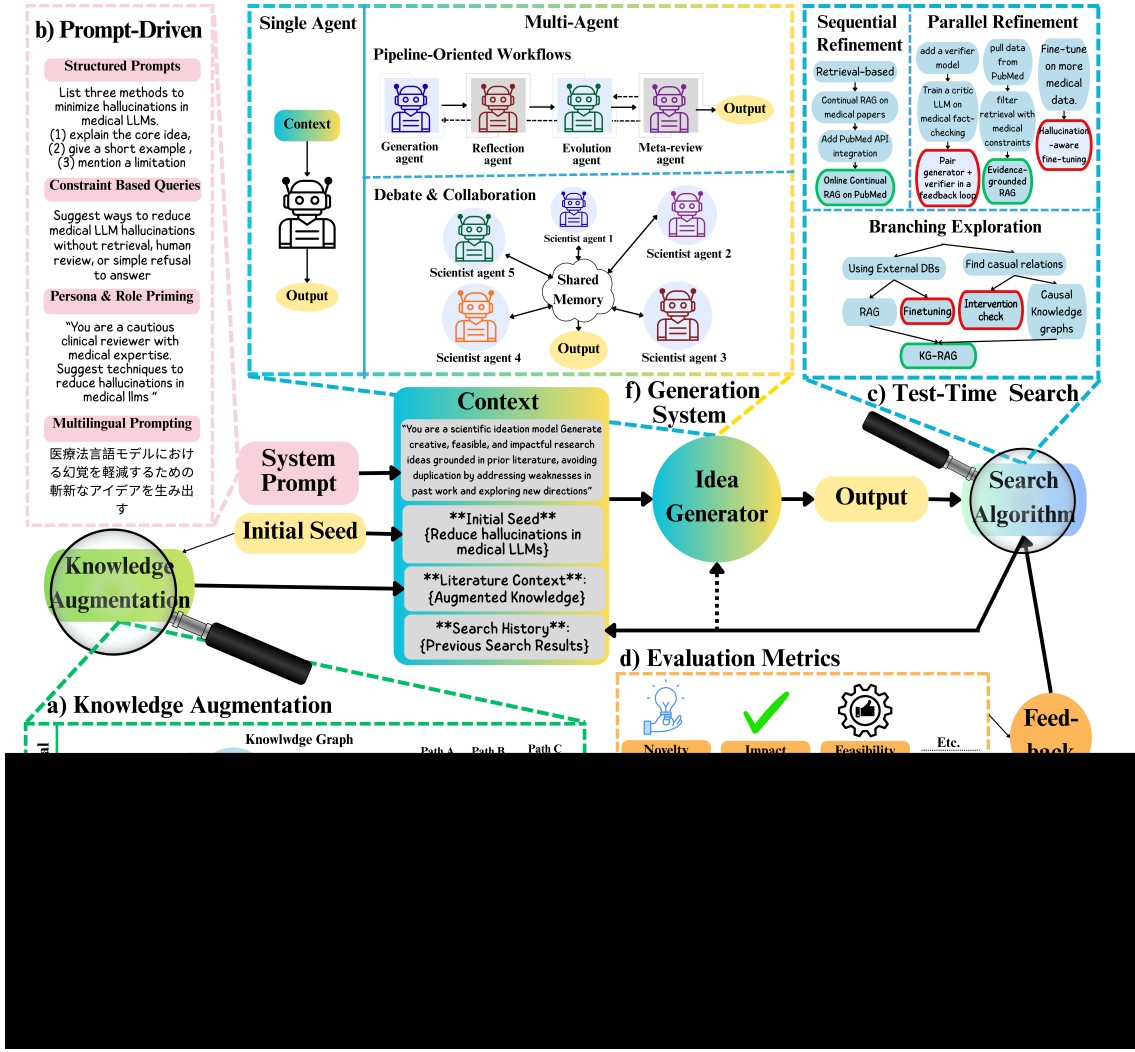

Figure 1: Overview of the training-free methods and requirements for scientific idea generation using LLMs. (a) *Knowledge augmentation:* The pipeline begins with integrating external sources such as research papers, databases, and knowledge graphs to ground the model's reasoning, either through relational linking or semantic similarity-based retrieval. (b) *Prompt-driven techniques:* The model can be steered by modifying system prompts or input instructions, including structured prompts, adversarial queries, persona and role priming, and multilingual prompting. (c) *Test-time scaling via search:* At inference time, search-based methods explore multiple candidate ideas through sequential refinement, parallel refinement, or branching exploration, enabling scalable reasoning. (d) *Evaluation metrics:* Candidate hypotheses are evaluated based on key scientific metrics such as novelty, feasibility, and potential impact. (e) *Feedback sources:* Search and generation are guided by feedback signals from human experts, scientific rules, internal model confidence, or external tools/simulators, which iteratively refine and improve the generated ideas. (f) *Generation system:* Single-agent systems versus multi-agent frameworks. Multi-agent setups include pipeline-oriented workflows for automation and debate-based interactions, which can yield emergent behaviors in LLMs. This multi-stage process collectively enables LLMs to produce creative, reliable, and high-value scientific hypotheses.

## 2 Knowledge Augmentation: Grounding LLMs in External Evidence

Scientific ideation rarely starts from a blank slate; researchers typically draw on prior literature, data, and domain knowledge to frame questions and generate hypotheses. In contrast, LLMs rely on a static knowledge base limited to their pretraining cutoff, motivating the need for access to up-to-date and domain-relevant information at inference time. Incorporating external literature or factual resources into the model's context grounds generation in established knowledge, reduces hallucinations, and serves as a source of creative input from the environment. From the perspective of Boden's taxonomy, knowledge augmentation primarily supports *combinatorial* creativity by enabling novel ideas through the recombination of existing concepts rather than transformation of the conceptual space. Indeed, Gu et al. demonstrate that retrieval-augmented models can explicitly enable combinatorial creativity by synthesizing research ideas across different abstraction levels. This mechanism aligns with the *press* dimension of Rhodes' 4Ps (Rhodes, 1961), where external context introduces novelty while preserving grounding and correctness. We survey two main approaches—*Semantic retrieval*, which dynamically incorporates evidence from external corpora, and *Relational retrieval*, which provides structured grounding—and discuss their limitations and open challenges. Figure 1(a) illustrates an example of knowledge augmentation with these two approaches.

### 2.1 Semantic Retrieval

Scientific ideation often begins by searching for nearby ideas in the existing literature. RAG systems extend LLMs with this capability: they query external sources (e.g., Semantic Scholar, EuropePMC, or vector databases) to retrieve relevant papers or text fragments based on semantic similarity. The retrieved content is then injected into the prompt, providing the LLM with context to enrich and constrain hypothesis generation.

Originally designed for open-domain question answering, RAG (Lewis et al., 2020) has become foundational in scientific idea generation. Early systems like *PaperQA* (Lála et al., 2023) and *LitLLM* (Agarwal et al., 2024) focused on raw literature access, while newer frameworks like *Ideasynth* (Pu et al., 2025), *Scideator* (Radensky et al., 2024), *Nova* (Hu et al., 2024), *ResearchAgent* (Baek et al., 2024)) emphasize multi-stage decomposition, critique-and-refine loops, and citation-aware retrieval. Furthermore, systems such as *SciPIP* (Wang et al., 2024c), *SCI-IDEA* (Keya et al., 2025), and *PaperHelper* (Yin et al., 2025a) push beyond simple matching by integrating novelty detection, visual scaffolding, and hybrid fine-tuning.

Across this line of work, retrieval is not merely about collecting documents—it is about curating and structuring the inspiration space. However, these methods remain rooted in similarity-driven access, which, while effective for relevance, can limit creativity by keeping exploration close to known clusters of knowledge.

### 2.2 Relational Retrieval

Similarity-based RAG excels at finding "neighbors" but scientific discovery often emerges from connecting distant or cross-disciplinary concepts. Knowledge Graphs (KGs) address this by encoding explicit relations among entities—concepts, methods, results, or papers—forming a structured topology for exploration.

KG-based approaches allow LLMs to traverse paths across domains, promoting more creative and meta-level reasoning than local similarity alone. In biomedicine, systems like *PubTator* (Wei et al., 2013; 2024) and *SciSpacy* (Neumann et al., 2019) extract subgraphs from literature to enable link prediction and hypothesis generation.

An early example of Relational Retrieval is *Chain of Ideas* (Li et al., 2024a) that by explicitly tracking breakthroughs and their dependencies, enabled LLMs to reason over trends rather than isolated documents. This inspired subsequent work that generalized the idea to broader graph formalisms. The *KG-CoI* framework (Xiong et al., 2024), *SciMON* (Wang et al., 2024a) and *Grounding LLM Reasoning with Knowledge Graphs* (Amayuelas et al., 2025) demonstrate that graph-structured multi-hop reasoning improves plausibility and interpretability. Large-scale efforts such as *Graph of AI Ideas (GoAI)* (Gao et al., 2025b) and *SciMuse* (Gu & Krenn, 2025) show how domain-spanning graphs can surface trends, gaps, and opportunities invisible to similarity-based retrieval.

In short, KG approaches ground LLM reasoning and expand LLM ideation in relational rather than purely semantic space. By traversing structured dependencies and cross-domain links, they open the door to more multidisciplinary and potentially more creative hypothesis generation.

## 2.3 Takeaways and Limitations

RAG-style and graph-based retrieval answer complementary needs in LLM-assisted scientific ideation. As shown in Scenario 2[1], Similarity-driven RAG gives fast, relevant context — it pulls the nearest prior work and facts so the model can be accurate and up-to-date. Relational retrieval, by contrast, exposes structure and cross-domain paths that are invisible to nearest-neighbour search, enabling multi-hop, meta-level, and multidisciplinary leaps.

Empirical evidence from GoAI study— which explicitly organizes and traverses ideas using a structured graph— (Gao et al., 2025b) shows that GoAI achieves higher Elo-based novelty scores (3.83) and the highest overall average score (3.39) compared to less structured baselines, such as Chain-of-Ideas(Li et al., 2024a) (3.21 novelty, 3.27 average) and vanilla LLM prompting (2.44 novelty, 2.70 average). These results indicate that relational structuring of ideas does more than improve relevance or clarity: it supports combinatorial creativity by enabling non-trivial recombination across ideas, while maintaining feasibility and effectiveness 3.

Despite these advances, heavy reliance on retrieved content can bias ideas toward well-represented clusters, reinforcing conservative or incremental thinking rather than encouraging novelty. Empirical studies confirm these tendencies: higher retrieval quality often reduces output diversity as models **overfit** to retrieved passages (Lee et al., 2024); decoding strategies overweight context at the expense of parametric reasoning (Wang et al., 2025b); and when external evidence conflicts with internal knowledge, LLMs exhibit confirmation bias rather than creative divergence (Lee et al., 2025).

Addressing these limitations may require hybrid strategies that balance grounding with exploration, incorporate richer modalities, and explicitly encourage divergence—directions we revisit in subsequent sections on prompt-based creativity and inference-time scaling.

---

[1]To aid intuition, we also include *scenario boxes* throughout this survey. These illustrative examples are **synthetic**—not actual system outputs—but are designed to clarify how different methods might operate in practice. By grounding abstract mechanisms in concrete cases, the boxes highlight contrasts between approaches and help readers better visualize potential reasoning and generative steps, without implying empirical results.

**Scenario: Similarity-driven vs. Relation-driven retrieval**

**Seed research goal:** Design methods to make LLMs more reliable when solving reasoning tasks in out-of-distribution (OOD) settings.

**Semantic-driven Retrieval.** The system generates a query such as "robustness in LLM reasoning under OOD" and retrieves the top-$k$ semantically similar papers from a vector database or literature API.
*Illustrative retrieved items:*

- Paper A: Chain-of-thought prompting improves reasoning in arithmetic tasks.

- Paper B: Self-consistency decoding reduces reasoning errors in LLMs.

- Paper C: Fine-tuning LLMs with synthetic reasoning chains.

*Generated hypothesis:* Combine chain-of-thought prompting with self-consistency decoding and fine-tune on synthetic reasoning datasets to improve robustness on novel tasks.

**Relation-driven Retrieval.** The seed is mapped to nodes such as "LLM reasoning" and "distribution shift" within a knowledge graph of ML research concepts and papers. The system performs multi-hop traversal to find relational paths across subfields.
*Illustrative graph paths:*

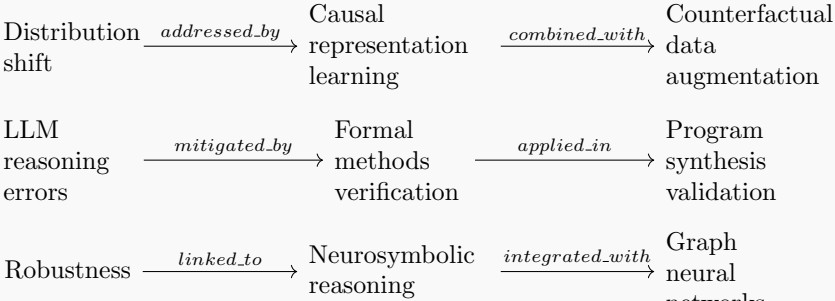

*Generated hypothesis:* Develop a hybrid framework where causal representation learning identifies shift-invariant features, augmented by counterfactual data generation, and pair it with neurosymbolic reasoning modules to improve LLM reasoning robustness under OOD conditions.

Similarity-driven retrieval emphasizes **local relevance** by surfacing closely related techniques, while relational retrieval encourages **cross-paradigm creativity**, uncovering connections unlikely to emerge from similarity alone. Despite these differences, standard *greedy decoding* in LLMs generally limits outputs from either approach to the combinatorial level.

Figure 2: Comparison of similarity-driven and relation-driven retrieval methods.

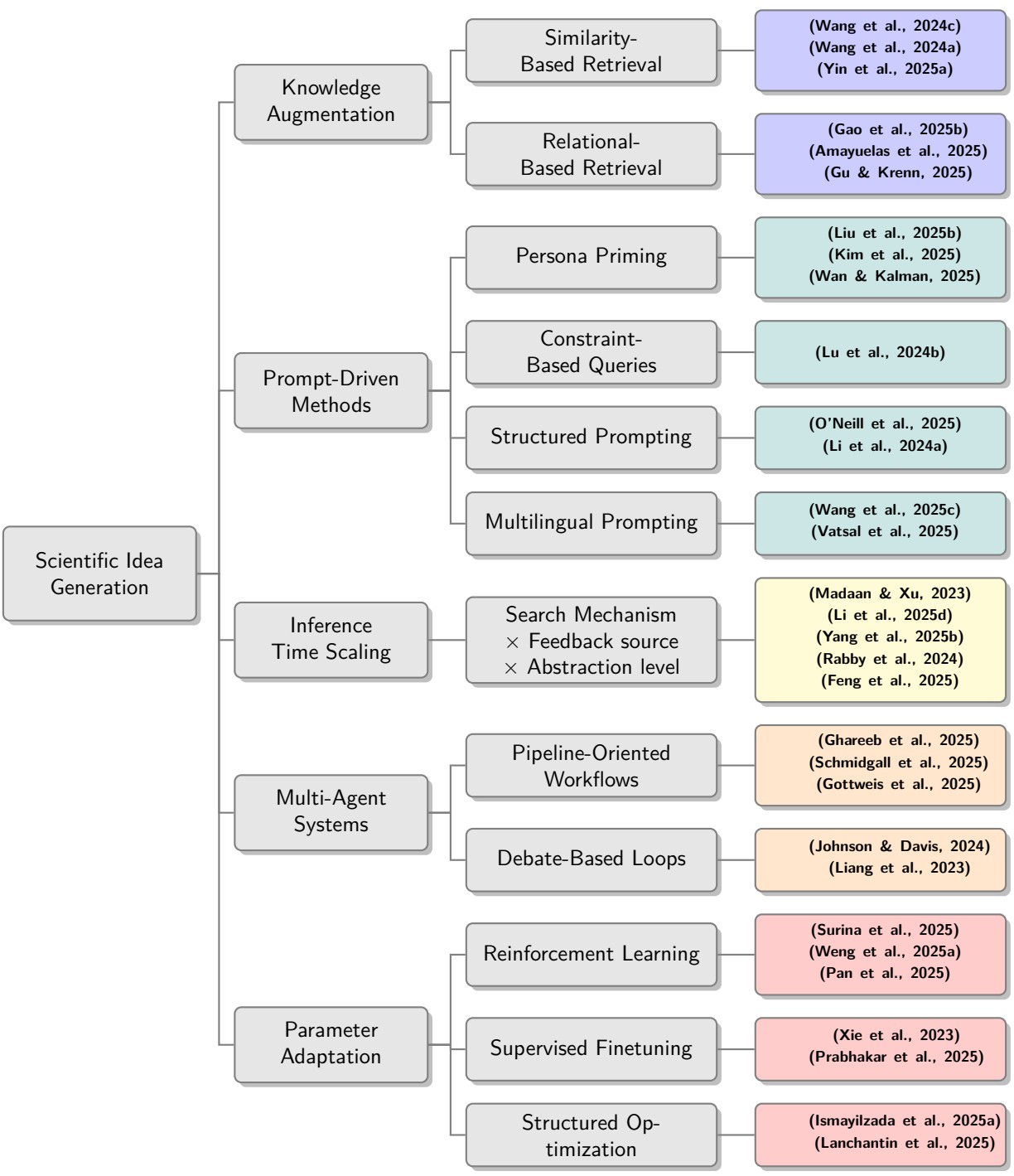

Figure 3: A five-level taxonomy for scientific idea generation methods using LLMs.

# 3 Prompt-Driven Creativity: Steering LLMs Towards Novel Ideas

Prompt engineering is a powerful and efficient method to influence LLMs without costly retraining or fine-tuning. By designing precise instructions, role contexts, or structured templates within the input prompt, researchers can steer LLMs toward generating novel ideas and creative solutions (Vatsal & Dubey, 2024). As prompts function as external constraints and affordances that shape the model's generative behavior, prompt driven methods align with the Press dimension of the 4P model trying to foster novelty while maintaining grounding in the model's pretrained knowledge. Figure 1(b) schematically depicts these prompt-steering strategies and their placement within the idea-generation workflow. Building on this view, we discuss four main prompt-driven techniques for diversifying outputs: *Persona & Role Priming*, *Constraint-Based & Adversarial Queries*, *Structured Creative Prompts*, and *Multilingual Prompting*.

## 3.1 Persona & Role Priming

Assigning expert identities or roles to LLMs helps sharpen creative output. Zhao et al. (2025) showed that role-based prompts—such as instructing the model to "act as a scientist"—can boost originality significantly on psychometric creativity tests. Supporting this, recent work in 2025 proposes frameworks that enhance LLM persona-awareness over multiple dialogue sessions, improving consistency and contextual relevance in outputs (Liu et al., 2025b). Another study explores how persona conditioning influences model alignment and ethical decision-making, highlighting the importance of carefully designed persona prompts to guide LLM creativity while avoiding unintended biases (Kim et al., 2025). Moreover, studies on human-AI co-creativity show that even ambiguous or counterintuitive personas can spur unexpected ideas, suggesting that nuanced role priming strategies can be highly effective in scientific ideation (Wan & Kalman, 2025).

## 3.2 Constraint-Based & Adversarial Queries

Prompting LLMs with constraints or adversarial instructions encourages them to explore less obvious, more innovative solutions. For example, asking models to challenge core scientific assumptions or forbidding common coding constructs can drive originality. Recent work introduces *Denial Prompting*, a method that incrementally imposes constraints on generated code to push models toward creative strategies (Lu et al., 2024b). This approach is evaluated using a new creativity metric called NeoGauge, demonstrating that systematically denying routine solutions helps LLMs access novel areas of their generative space (Lu et al., 2024b). These constraint-based methods effectively leverage the vast internal knowledge of LLMs while navigating the lower-probability, more inventive regions of output distributions.

## 3.3 Structured Creative Prompts

Structured prompts scaffold LLM reasoning, fostering more deliberate and diverse ideation. By guiding models through defined reasoning steps, they emulate human metacognitive strategies that enhance creativity. *Chain-of-Thought (CoT)* prompting, where models are encouraged to "think step by step," improves reasoning accuracy and idea diversity (Wei et al., 2022). Building on this, recent advances like *Consistency-Based Self-Adaptive Prompting (COSP)* automate zero-shot

prompt construction by leveraging the model's own predictions and unlabeled data, enhancing performance on reasoning tasks without task-specific training (Wan et al., 2023).

In scientific domains, methods like *Chain-of-Ideas (CoI)*, structures ideation by organizing literature into sequential knowledge chains that reflect the evolution of a research domain (Li et al., 2024a). Instead of exposing LLMs to unstructured data, CoI captures dependencies between past findings and emerging trends, enabling the model to identify gaps, trace idea development, and forecast future research directions. Likewise, *Bit-Flip-Spark*, structures idea generation into three stages: the *Bit* (baseline assumption), the *Flip* (challenging that assumption), and the *Spark* (novel insight that emerges) (O'Neill et al., 2025).

This scaffold is designed to induce cognitive conflict by prompting models to question default assumptions and adopt alternative premises, a mechanism commonly associated with creative ideation. Prior work shows that system-level instructions and structured reasoning scaffolds can reweight exploration and promote otherwise underrepresented trajectories within a fixed pretrained space (Reynolds & McDonell, 2021; Yao et al., 2023). In this more limited sense, structured prompting may be viewed as a procedural step toward transformational behavior, approximating aspects of conceptual space alteration through controlled search and assumption relaxation rather than true representational change.

---

**Scenario: When Prompting Alone Cannot Break Alignment Constraints**

**Task:** A researcher asks an LLM to *"Think outside the box and propose a completely new programming paradigm beyond object-oriented or functional programming."*

**Aligned LLM response (current state).** Despite being explicitly prompted to "be radically creative," the aligned model produces only **safe, incremental ideas**, such as:

- A hybrid paradigm combining object-oriented and functional programming.

- Event-driven programming with AI-assisted code generation.

- Domain-specific languages for scientific computing.

These outputs are plausible but unsurprising, reflecting the conservative bias from alignment training (e.g., RLHF).

**Unaligned / base model (hypothetical) response.** A less-aligned model might generate more radical and unconventional concepts, for example:

- A self-modifying, biology-inspired code ecosystem where programs evolve like neural tissue.

- A programming language treating time as a first-class dimension

These ideas may be impractical or incoherent, but they reflect **true divergent exploration** beyond the comfort zone of mainstream paradigms.

---

Figure 4: Comparison between aligned and unaligned LLM responses, highlighting how alignment training suppresses divergent creative exploration.

### 3.4    Multilingual Prompting

Recent studies reveal that multilingual prompting—translating prompts into multiple languages and aggregating the results—can significantly increase generative diversity and creativity. For instance, Wang et al. (2025c) found that cross-lingual prompting enables LLMs to draw on distinct linguistic and cultural priors, outperforming traditional diversity techniques such as temperature sampling. Similarly, a large-scale survey by Vatsal et al. (2025) across more than 250 languages highlights multilingual prompting as a broadly effective strategy for improving comprehension, reasoning, and creative output. The MLPrompt system further demonstrated how cross-lingual prompt reformulations can boost LLM comprehension and reasoning in complex tasks(Wang et al., 2025c). Together, these findings suggest that linguistic diversity itself acts as a cognitive scaffold—encouraging LLMs to traverse alternative semantic frames and produce more original, cross-cultural ideas.

### 3.5    Takeaways and Limitations

Prompt-based techniques—such as persona prompting, constraint based queries, structured, and multilingual prompting—can effectively broaden LLM creativity by diversifying perspectives, reasoning paths, and linguistic representations. Empirical results from Zhao et al. (2025), using a TTCT-inspired framework with LLM-judge scoring aggregated over many samples, show that prompt-based techniques systematically shift creativity components. Interpreting originality as novelty and the mean of fluency, flexibility, and elaboration as valueness, instructional prompting increases novelty from 3.78 to 4.05 and valueness from 4.46 to 4.65. Chain-of-Thought prompting primarily boosts valueness (4.75 vs. 4.68) with negligible change in novelty (3.76 vs. 3.78). Persona-based prompting (e.g., natural scientist) achieve higher novelty (4.23) and valueness (4.62) than generic roles (4.1, 3.78). Overall, prompt-based methods enhance creativity by redistributing generation within existing knowledge, with limited impact on deeper exploratory novelty.

However, as shown in Scenario 4 even with strong creativity cues in the prompt—e.g., "Be radically original," "Break all current conventions"— the aligned LLM remains trapped within a safe attractor basin created during RLHF and instruction tuning which prioritize safety and helpfulness over novelty. Some recent researches found that RLHF reduces output entropy and semantic variety, pushing models toward safer, less creative responses (Mohammadi, 2024; Kirk et al., 2023). Similarly, explicit creativity cues can sometimes reduce fluency and flexibility (Zhao et al., 2025). This shows why simply prompting for creativity is insufficient: the model's sampling distribution has been narrowed by alignment to avoid outputs that seem implausible or risky (Wolf et al., 2023). To address these limitations, inference-time scaling and multi-agent interaction offer a promising direction by expanding the search space beyond one-pass prompting, potentially enabling more novel, cross-disciplinary ideas (Feng et al., 2025; Baek et al., 2024).

# 4 Search and sampling expansions: Inference-Time Exploration of Hypothesis Spaces

Inference-time scaling methods expand the capabilities of LLMs beyond single-pass decoding by enabling dynamic and iterative exploration without additional training. From a classical AI perspective (Russell & Norvig, 2020), these approaches can be framed as search problems: LLMs act as agents traversing vast combinatorial spaces of candidate hypotheses, guided by heuristics and evaluation functions.

By moving beyond random sampling, test-time scaling highlights the role of *systematic search*, where both the *strategy of exploration* and the *abstraction level* of representation shape the creative potential of generated ideas. The search strategy—ranging from focused local refinement to tree-structured branching—determines the breadth and depth of traversal, while the abstraction level—spanning from low-level outputs to higher-level conceptual hypotheses—defines the scope of novelty that can emerge. We expect that applying more systematic search at more abstract representational levels increases the likelihood of producing *exploratory* ideas, aligning this process with Boden's notion of creativity as the structured traversal of a conceptual space (Boden, 2004).

Yet, exploration must be balanced against correctness and feasibility. While multi-sampling and branching strategies favor novelty and diversity, unconstrained search risks producing trivial or scientifically invalid ideas. To counter this, inference-time scaling methods incorporate *feedback signals* that guide pruning, refinement, and prioritization. In this way, feasibility and soundness are gradually reinforced without sacrificing exploratory breadth.

Overall, existing approaches can be analyzed along three complementary design dimensions. The **Search mechanism** defines the structure of exploration, such as local refinement, population-based search, or tree-structured branching. The **Feedback source** specifies the signals that shape traversal, ranging from self-consistency checks to empirical simulations and expert judgment. Finally, the node **Abstraction level** captures the granularity of ideas, from low-level token continuations to high-level hypotheses or executable workflows. These dimensions jointly determine how effectively inference-time scaling methods navigate the trade-off between novelty and feasibility, and they provide a principled framework for comparing strategies of scientific idea generation within the broader context of classical AI search and creativity theory.

## 4.1 Search Mechanism

Search mechanisms govern how LLMs traverse the hypothesis space during inference-time scaling for scientific ideation. In this section, we analyze three primary paradigms: **Local Search**, **Tree Search**, and **Population-Based Search**. Each paradigm embodies a distinct approach to expanding, evaluating, and selecting candidate scientific ideas, reflecting trade-offs between exploration, exploitation, memory cost, and computational efficiency. Figure 1(c) illustrates an example of idea expansion across various search mechanisms.

### 4.1.1 Local Search: Single-Candidate Refinement

Sequential refinement methods iteratively improve a single candidate hypothesis, analogous to *hill-climbing* in classical AI (Russell & Norvig, 2020). This approach mirrors the scientific practice of proposing an initial idea and then refining it through feedback cycles.

For example, the *Self-Refine* method (Madaan et al., 2023) uses internal feedback loops in which the LLM critiques its prior output and revises it without requiring additional training. *PANEL* (Li et al., 2025d) similarly decomposes reasoning into multiple steps, generating and critiquing each sequentially. *ResearchAgent* (Baek et al., 2024) extends this paradigm by simulating peer review with multiple collaborating agents, though the refinement of each hypothesis remains sequential. *CriticAL* (Li et al., 2024b) integrates statistical hypothesis testing into this loop, enabling the detection of logical inconsistencies or unsupported claims.

A single-path search, while simple and low-cost, is fundamentally brittle. It may terminate with a suboptimal or uninteresting solution if the initial path leads to a local maximum, making it an incomplete method that risks missing a solution entirely. In contrast, beam search method designed to overcome this limitation by exploring multiple paths concurrently. Scenario 5 demonstrates how different search strategies address this trade-off between efficiency and exploration.

### 4.1.2 Population-Based Search: Beam-Style Refinement

Population-based methods maintain multiple candidate hypotheses simultaneously, propagating the most promising across iterations. This aligns with *Local Beam Search*, a natural extension of single-candidate local search (Russell & Norvig, 2020). Maintaining a population allows models to balance exploration and exploitation, reduce the risk of local optima, and improve diversity.

Modern LLM examples include *MOOSE-Chem2* (Yang et al., 2025b), which run multiple instances or seeds in parallel, scoring candidates with internal or external evaluators. These approaches leverage ensemble diversity while maintaining refinement dynamics akin to local search.

Although local beam search reduces the brittleness of single-path refinement by maintaining a population of candidates, it remains constrained: once a branch is discarded, it cannot be recovered (Scenario 5). This means the method may still miss promising but initially under-performing directions. To address this limitation, researchers turn to tree-based approaches that allow a more systematic exploration of alternatives.

### 4.1.3 Tree Search: Branching Exploration

Branching exploration maps closely to *tree-based informed search*, where instead of tracking only a fixed-width beam, the search process can expand, backtrack, and revisit multiple paths, guided by heuristics that balance depth and breadth. This structured exploration mitigates premature pruning and supports the discovery of more diverse solutions. (Russell & Norvig, 2020).

*MC-NEST* (Rabby et al., 2024) exemplifies a Monte Carlo Tree Search guided by Nash equilibrium principles, balancing exploration and exploitation through stochastic node selection strategies. Conceptually, this approach mirrors classical local search techniques such as *stochastic hill climbing* and *simulated annealing* (Russell & Norvig, 2020), in which the search tolerates occasional suboptimal moves to escape local minima. *MAGIC* (Xu et al., 2023a) applies multi-armed bandit strategies to allocate computational resources preferentially to promising branches. *Monte Carlo Thought Search* (Sprueill et al., 2023) explores alternative synthesis pathways in catalysis design. Multi-agent systems, such as *Virtual Scientists* (Johnson & Davis, 2024) and the *IRIS* framework (Feng et al., 2025), implement branching exploration collaboratively, integrating both LLM and human feedback to prune and guide search.

---

**Scenario:   Search Mechanisms in Hallucination Mitigation**

**Seed Idea:** "Design a method to reduce llm factual hallucinations in medical questions."

**1. Local Search (Sequential Refinement)**: The model iteratively refines a single idea;
   **Iteration 1:**   Add RAG with PubMed.
   **Iteration 2:**   Integrate retrieval into every step of multi-hop reasoning.
   **Iteration 3:**   Combine retrieval with a confidence estimator; rejects uncertain answers.
Focuses on refining a single path, which may lock the model into one paradigm (e.g., RAG)
and overlook other promising strategies like calibration or verification.

**2. Local Beam Search (Parallel Refinement)**: Multiple candidate directions are explored simultaneously;

- **RAG-based factual grounding** → iteration 1 → iteration 2 → ...

- **Uncertainty calibration methods** → iteration 1 → iteration 2 → ...

- **Training with hallucination-focused preference data** → iteration 1 → ...

After each refinement round, weaker ideas (e.g., costly preference data) are dropped, while
stronger ones continue. Balances multiple strategies before committing, but innovative yet
noisy ideas may be pruned prematurely.

**3. Tree Search (Branching Exploration)**: Structured hierarchical exploration of the
idea space;

- **Branch A: Retrieval-based approaches**

  - Hybrid BM25 + Dense retrievers
  - Domain-specific knowledge bases

- **Branch B: Training-time alignment**

  - Hallucination-specific RLHF
  - Adversarial training with hallucination prompts

- **Branch C: Uncertainty-aware generation**

  - Selective abstention
  - Calibrated probability outputs

Heuristics like computational cost determine branch expansion, Preserve diverse strategies
and enables backtracking, though strong heuristics are required to manage combinatorial
growth.

---

Figure 5: Comparison of sequential, parallel, and tree-based search mechanisms for the ideation
task of reducing LLM hallucinations in medical question answering.

Across these mechanisms, we see a spectrum of trade-offs in how LLMs explore idea space during inference-time scaling. As shown in Scenario 5, single candidate refinement offers simplicity and efficiency but risks tunnel vision, converging too quickly on a narrow line of thought. Beam-Style refinement introduces diversity by allowing multiple candidates to coevolve, yet its pruning decisions are irreversible and may discard unconventional but valuable paths (Zhou & Hansen, 2005; Meister et al., 2020; Cohen & Beck, 2019). Tree search provides the broadest and most systematic exploration, preserving multiple paradigms in parallel -though its ability to preserve unconventional hypotheses depends on the choice of heuristics- and enabling backtracking, a mechanism shown to be critical for recovering from premature commitments in recent LLM reasoning frameworks (Yao et al., 2023), albeit at significant computational cost.

Together, these methods highlight that no search strategy alone guarantees optimal idea generation. Greater systematic exploration—approaching tree-like search—tends to increase the creativity level of idea into exploratory in boden framework, but this comes at the cost of efficiency. Their effectiveness, however, depends on the reward source that guide search and defines how candidate hypotheses are prioritized and refined —whether from internal confidence, external evaluators, or environment feedback- so we next examine the role of *feedback sources* in scientific ideation.

### 4.2 Feedback Source

Feedback sources in inference-time scaling act as evaluators that shape the trajectory of idea exploration. While most commonly used to verify feasibility, soundness, and scientific grounding, they can also assess novelty and creativity by identifying underexplored branches or highlighting more promising directions. Crucially, each feedback source has strengths in particular domains. **Model confidence** signals are effective in relatively simple settings, where the model's general knowledge suffices to infer correctness. **Peer evaluator** agents offer greater flexibility and domain specialization, though they remain vulnerable to reward hacking. **Domain rules** and symbolic constraints are most effective in well-formalized fields, where correctness is clearly defined. Finally, **Human evaluation** is especially valuable in subjective domains where intuition, creativity, and contextual judgment are essential. Together, these complementary feedback sources provide the necessary guidance to balance exploratory breadth with scientific validity in inference-time scaling.

#### 4.2.1 Internal Self-Evaluation

A foundational feedback source is the model's internal confidence, enabling self-critique and refinement without external input. Methods like *Self-Refine* (Madaan et al., 2023) leverage the LLM's ability to generate critical commentary on its own outputs and revise accordingly. *PANEL* (Li et al., 2025d) decomposes reasoning into discrete steps, each followed by model-generated critiques. *CriticAL* (Li et al., 2024b) further enhances this by applying statistical reasoning—such as hypothesis testing—to identify contradictions and weaknesses.

It is most effective when the model's pretrained general linguistic knowledge suffices (e.g., logical consistency, surface fluency). However, numerous studies show that LLM self-confidence is often miscalibrated: models can be markedly overconfident in incorrect outputs or under-confident in correct ones, with accuracy–confidence alignment heavily influenced by prompt framing, distractors, and training objectives (Geng et al., 2024; Leng et al., 2025; Chhikara, 2025). As a result, reliance on internal heuristics risks *bias reinforcement* and even a form of *reward hacking*, where the model over-trusts its own scoring signals, strengthens spurious associations, or repeatedly exploits flaws in

its confidence estimates without genuinely improving hypothesis quality. n Scenario 6, we compare the effects of internal confidence feedback versus peer-agent feedback on hypothesis evaluation.

---

**Scenario: Internal Confidence vs Peer Agents in Hypothesis Evaluation**

Suppose an LLM proposes a novel compound, $C_{22}H_{28}N_6O_4$, as a potential inhibitor for a cancer-related enzyme.

**Internal Confidence Feedback.** The model rates the compound as valid because its structure "looks" consistent with patterns learned during pretraining. It deems the structure plausible because it resembles valid chemical formulas and mimics motifs frequently seen in known inhibitors.

**Peer-Agent Feedback.** A domain-specialized peer agent reviews the same compound and identifies structural instabilities or unrealistic bond angles that make it chemically unfeasible and flags it as unstable.

Internal confidence is efficient and scalable, helping flag syntactically invalid outputs, but it may overestimate validity by focusing on surface-level coherence rather than underlying chemical feasibility. While peer-agent feedback enhances scientific rigor and domain awareness, though it requires additional computational and training resources to maintain reliable expert peers.

---

Figure 6: Contrasting internal confident and peer-based feedback in scientific hypothesis evaluation.

### 4.2.2 Peer LLMs as Verifiers

A growing line of work leverages LLMs as *external evaluators*, distinct from the generator itself. For example, *FlexiVe* (Zhong et al., 2025) implements a solve–detect–verify pipeline, while process-reward frameworks (Choudhury, 2025) and diverse verifier tree search use verifier models to score intermediate hypotheses and control expansions. Stability can be improved by ensemble verification (Wang et al., 2025a).

The *CRPO* (Ismayilzada et al., 2025a) exemplifies this approach by using an LLM's general linguistic knowledge as a feedback signal for guiding logical consistency and language fluency. Similarly, the *LLM-as-a-Judge* (Zheng et al., 2023) paradigm formalizes this idea: one model (or ensemble (Chen et al., 2025a)) evaluates hypotheses produced by another, effectively operationalizing externalized language-model judgment as search heuristics.

More advanced methods first train specialized evaluators to capture problem-specific criteria before deploying them as verifiers. For example, *CycleResearcher* (Weng et al., 2025a) relies on a trained verifier model (CycleReviewer) to judge intermediate steps and direct search trajectories. Li et al. (2024c) similarly build a reward dataset and trains three reward models to guide idea generation dynamically, allowing fine-grained control over different evaluation dimensions. These approaches are often positioned as scalable, automated substitutes for human feedback, especially in domains

where expert evaluation is costly. Peer evaluators are more flexible than self-confidence, since they may be domain-specialized and provide rationales.

However, they remain vulnerable to reward hacking: generators can learn to exploit systematic weaknesses in verifier heuristics, producing outputs that appear highly rated without being genuinely correct or useful. Recent studies in preference optimization and RLHF have shown that models can overfit to their evaluators, generating verbose, sycophantic, or superficially persuasive responses that maximize reward signals rather than substantive quality (Laidlaw et al., 2025; Miao et al., 2024). These findings highlight that while peer evaluators extend the evaluative horizon, their reliability depends on careful design to avoid exploitation and collapse of evaluation diversity. While effective as evaluators and filters, feedback sources differ fundamentally from collaborative multi-agent setups. As outlined in Section 5, *multi-agent frameworks* engage additional LLMs as **co-creative partners**, enabling them not only to assess but also to propose, contest, and refine ideas in a shared search process.

### 4.2.3 Simulators and Real Environments

Simulators serve as semi-grounded environments that test the outputs of LLMs in dynamic, feedback-rich contexts. These can be physical simulations, digital twins, or even agent-based societies. Simulators and real environments provide feedback through interactive trial-and-error, aligning closely with notion of *online search*(Russell & Norvig, 2020). Unlike offline planning, where a complete solution is computed in advance, online search emphasizes interleaving generation with execution: the agent proposes an action, observes the environment's response, and adapts its trajectory accordingly. This makes simulators and lab-in-the-loop systems natural analogues, as they allow LLM-driven agents to iteratively refine hypotheses or strategies based on grounded feedback from physical laws, empirical trials, or synthetic environments.

*Robin* (Ghareeb et al., 2025) uses a "lab-in-the-loop" design, where hypotheses are tested via in silico simulators (or even robotic wet-labs), and the resulting outputs are looped back into the ideation process as reward signals. For example, a gene-editing strategy might be generated, simulated for efficacy, and then refined based on empirical response.

Similarly, *AtomAgents* (Ghafarollahi & Buehler, 2024) combines LLMs with physics-based alloy simulators: agents propose material compositions, query simulators for predicted properties like tensile strength, and prioritize candidates for further iteration based on performance metrics.

In more abstract contexts, *CASEVO* (Jiang et al., 2024) simulates agent societies debating political or scientific stances, tracking how persuasive interactions shift beliefs—modeling ideation in socially complex domains.

Recent robotics work demonstrates the coupling of simulators and real execution. *SayCan* (Ahn et al., 2022) grounds LLM-generated high-level commands in an affordance model and low-level robot controller, combining planning tools with real robot feedback. *RoboTool* (Xu et al., 2023b) extends this to creative tool use, where LLMs propose robot code tested in both simulators and physical hardware. *Voyager* (Wang et al., 2023a) highlights how open-ended exploration in Minecraft enables the synthesis of reusable tools, iteratively evaluated in a simulator to accumulate a growing library of skills.

On the evaluation side, Torne et al. (2024) analyze the fidelity of simulator-to-reality transfer, explicitly comparing robot manipulation policies in simulation versus physical execution. Similarly, Large-Scale Simulation of LLM-Driven *Generative Agents* stress-tests agentic LLMs in synthetic societies, showing how simulation feedback can approximate large-scale environment signals for long-running scientific ideation (Piao et al., 2025). As shown in the Scenario 7, the simulator evaluates stability and feasibility of candidate designs.

### 4.2.4 External Tools and APIs

External tools expand the feedback horizon by integrating APIs, online search, and domain-specific databases. The *Toolformer* architecture (Schick et al., 2023) equips LLMs with the capability to query external resources dynamically, enabling verification and augmentation of generated content. Chemical synthesis platforms like *ChemCrow* (Bran et al., 2023) leverage cheminformatics tools and external data sources to validate reaction pathways proposed by the model. Similarly, *HuggingGPT* (Shen et al., 2023) coordinates a wide range of specialized AI models hosted on the Hugging Face platform, allowing an LLM to act as a controller that routes tasks such as vision, speech, and text analysis to appropriate external tools, demonstrating how tool integration can expand an LLM's capabilities beyond text-only reasoning.

Recent evaluations focus directly on LLM–tool interaction. *MINT* (Wang et al., 2024d) systematically studies multi-turn tool usage, showing how feedback from tool calls and natural language guidance improves decision-making. Voyager's open-ended tool synthesis also bridges this category, where tools (skills, functions) are generated internally but then tested through simulators and APIs.

By incorporating external tools, LLMs move beyond isolated language generation toward grounded, multi-modal reasoning informed by up-to-date, external knowledge. This shift transforms them from pure text predictors into decision-theoretic agents augmented with specialized oracles. This feedback source and its alternatives are examined in Scenario 7.

### 4.2.5 Scientific Priors and Domain Rules

Incorporating feedback from hard-coded or learned scientific priors enables models to remain within physically or logically valid domains. These systems operate without full simulation but leverage symbolic constraints or heuristic checklists to prune the hypothesis space. This connects closely to discussion of *constraint satisfaction problems*, where domain knowledge narrows search to feasible regions.

*SCaSML* (Fan et al., 2025) integrates physical laws such as energy conservation to constrain generation, ensuring outputs remain within scientifically plausible bounds without requiring explicit simulation feedback. Similarly, methods like $R^3V$ (Cheng et al., 2024) embed domain logic checkpoints to detect and correct illogical reasoning mid-generation. The Scenario 7 illustrates how scientific priors enforce fundamental constraints and ensure theoretical validity. Domain rules are most effective in well-formalized fields (e.g., physics, chemistry, mathematics) where symbolic constraints or laws are clear and non-negotiable. Their strength is reliability: they are resistant to reward hacking and scale well. However, their rigidity limits flexibility in ill-defined or emergent domains, where strict rules may exclude innovative or unconventional ideas.

### 4.2.6 Human-in-the-Loop Evaluation

Human experts remain the gold standard for nuanced and context-aware evaluation. Unlike automated evaluators, humans can flexibly weigh novelty, relevance, and feasibility in ways not easily reducible to rules or simulations.

Platforms like (He et al., 2024) integrate human expertise directly into the scientific discovery process by allowing researchers to iteratively guide and critique model-generated hypotheses. *IdeaSynth* (Pu et al., 2025) and *Scideator* (Radensky et al., 2024) provide interactive environments for hypothesis refinement, allowing researchers to blend and adapt elements from prior work. *Chain-of-Table* (Wang et al., 2024e) offers a mixed-initiative framework where human evaluation is combined with structured scientific tables and knowledge bases, balancing creativity with formal rigor. *IRIS* (Feng et al., 2025), though originally designed for self-reflection, also incorporates human evaluators in tasks requiring subjective judgment of novelty and coherence. These human-in-the-loop interactions enrich the feedback loop, ensuring that the discovery process not only leverages computational power for exploration but also incorporates experiential insights that are difficult to encode explicitly, ultimately leading to more reliable and interpretable scientific advancements.

---

**Scenario : Contrasting Evaluation feedback sources**

To illustrate how different feedback sources shape ideation, consider an LLM that proposes a novel photo-catalyst for converting $CO_2$ into useful chemicals under sunlight. The idea is passed through several evaluators, each providing a distinct form of guidance:

- **Simulator:** The design is tested in a simulation engine, which shows promising conversion rates but also reveals that the material becomes unstable at high temperatures. The simulator narrows the search space to only variants that remain physically viable.

- **External database:** The LLM queries scientific databases and patents. It finds no direct precedent for the exact design (a sign of novelty), but flags that similar catalysts once produced toxic byproducts. This feedback shifts exploration toward safer material choices.

- **Scientific priors:** Symbolic scientific rules check the proposed reaction. They detect a violation of basic energy conservation, forcing the model to reformulate the mechanism in a physically consistent way. Priors act as hard constraints that keep the search scientifically valid.

- **Human expert:** A human expert reviews the surviving ideas. Beyond technical feasibility, she considers manufacturability, cost, and environmental impact, and directs the system toward designs using abundant, scalable materials.

Together, these perspectives highlight how different feedback sources contribute complementary strengths: simulators ground ideas in empirical behavior, external tools provide contextual knowledge, priors ensure theoretical soundness, and humans add judgment and broader vision.

---

Figure 7: Contrasting evaluation feedback sources for LLM-driven scientific ideation.

Human-in-the-loop evaluation is especially valuable in *subjective* or *poorly formalized* domains (e.g., social sciences, exploratory biology) where intuition, creativity, and context matter. It provides robustness against reward hacking and fills gaps left by incomplete simulations or priors. However, it is costly, time-consuming, and hard to scale—limiting its use in high-throughput or automated scientific ideation pipelines.

### 4.3 Levels of Abstraction

Inference-time scaling methods differ not only in their strategies but also in the *level of abstraction* at which search is conducted. From a theoretical perspective, abstraction provides a means of reducing or restructuring the problem space, thereby enabling broader exploration but complicating verification (Zucker, 2003; Nakar et al., 2024). Recent LLM-based methods explicitly operationalize this trade-off. For instance, Wang et al. (2024b) show that separating hypothesis generation at an abstract level from concrete program instantiation improves performance on reasoning tasks, while Singhal & Shroff (2024) demonstrate that concept-level scoring functions allow more efficient traversal of search spaces than surface-level metrics. Together, these studies suggest that higher levels of abstraction broaden the search space and bring the process closer to exploratory creativity, but at the cost of increased difficulty in assessing feasibility and correctness.

In scientific idea generation field, some approaches direct LLMs toward ideating high-level scientific **Hypotheses**, while others instruct them to produce executable **Program** or simulation code. A further class operates at the **Meta-level**, searching over the model's own instruction or modes of interaction with the environment.

#### 4.3.1 Hypothesis-Level Ideation

At the highest level of abstraction, models function as idea generators. They propose conceptual hypotheses, causal explanations, or theoretical insights without grounding them in execution or empirical validation. The focus lies in novelty, coherence, and domain relevance, supporting early-stage brainstorming or hypothesis formulation.

For instance, *MAGIC* (Xu et al., 2023a) generates speculative hypotheses in genomics and systems biology, ranking them by plausibility and novelty heuristics. *VirSci* (Johnson & Davis, 2024) applies multi-agent debate to refine hypotheses through peer critique. *IRIS* (Feng et al., 2025) adds internal reflection, restructuring ideas mid-generation to avoid incoherence or excessive derivativeness. These systems act as cognitive companions: their outputs are inspirations requiring downstream verification by humans or automated evaluators. An example of hypothesis-level ideation is illustrated in Scenario 8.

#### 4.3.2 Program-Level Ideation

A second class of methods situates search at the programmatic layer, where models not only propose *what* to test but also *how* to test it. Here, LLMs generate structured protocols—pseudocode, experiment scripts, or API call sequences—that can be executed by simulators, databases, or robotic systems. The results are then looped back, closing the gap between hypothesis generation and empirical evaluation.

---

**Scenario: Hypothesis-Level Ideation (Mathematical Reasoning)**

An LLM proposes the conjecture:

*"For any odd prime p, the sum of quadratic residues modulo p equals $\frac{p(p-1)}{4}$."*

This is a purely abstract claim without proof. The model explores nearby conjectures (e.g., cubic residues, alternate modulus conditions). Evaluation requires external proof attempts or human mathematicians.

**Search style:** Local search over conceptual variants of conjectures.

---

Figure 8: Illustration of hypothesis-level ideation in mathematical reasoning.

*Program-Aided Language models (PAL)* (Gao et al., 2023) exemplify this principle by translating natural-language reasoning into Python programs executed externally, outperforming direct chain-of-thought reasoning.*HuggingGPT* (Shen et al., 2023), *Toolformer* (Schick et al., 2023), and Explanations as Programs (Zhang & Huang, 2023) similarly teach models to output modular, debuggable tool pipelines. In science, *LLM-SR* (Shojaee et al., 2024) refines symbolic equations by querying solvers for constraint violations, *ChemCrow* (Bran et al., 2023) plans synthesis routes via cheminformatics APIs, and *CodeScientist* (Huang & Wu, 2024) generates executable lab protocols for simulators or robotic wet-labs.

Ideation at the program level enables the use of simulators and other external tools by integrating execution directly into the generative loop. An example of program-level ideation is illustrated in Scenario 9. This integration offers key benefits: programs are structured and interpretable, their execution provides robustness against unsupported claims, and hallucination is reduced as hypotheses must be empirically validated through execution.

---

**Scenario: Program-Level Ideation (Mathematical Reasoning)**

Instead of stating a conjecture, the LLM outputs a structured proof plan:
1. Represent quadratic residues as $a^2 \bmod p$.
2. Pair each $a^2$ with $(p-a)^2$.
3. Show these pairs sum to $p$.
4. Derive a closed-form expression for the total sum.

Each step can be checked in a proof assistant or tested computationally.

**Search style:** Tree or beam search, where alternative proof strategies (induction vs. direct algebraic argument) form different branches.

---

Figure 9: Illustration of program-level ideation in mathematical reasoning.

### 4.3.3 Meta-Level Search and Self-Adaptation

A third, emerging frontier is meta-level search, where the object of optimization is not the hypothesis or program but the agent's own *configuration and interaction* with its environment. Here, systems explore changes to their prompting strategies, exploration parameters, reward weightings, or even architectural components, adapting the search process itself. An example of meta-level ideation is illustrated in Scenario 10.

The *Darwin Gödel Machine* (Zhang et al., 2025a) exemplifies self-modification: instead of emitting domain-level programs, it evolves its own reasoning code to improve performance. More broadly, meta-search also includes adaptive inference strategies—e.g., adjusting beam widths, modifying exploration–exploitation trade-offs, or re-balancing between simulator calls and heuristic priors. Such mechanisms parallel meta-reasoning and utility-based agent adaptation, where the agent learns not only *how to act*, but *how to search*. Another notable example is the Meta-review agent in Google's *AI co-scientist* system (Gottweis et al., 2025). This agent synthesizes feedback from various agents, identifies recurring issues, and generates comprehensive critiques and research overviews. The Meta-review agent compiles feedback across iterations to adjust the agents' prompts, ensuring continual improvement.

Meta-level search completes the abstraction spectrum: from generating hypotheses, to encoding them in executable plans, to adapting the very rules by which hypotheses and programs are generated. This higher-order flexibility may prove critical for scaling LLMs into autonomous scientific explorers.

---

**Scenario: Meta-Level Ideation (Mathematical Reasoning)**

The LLM reconfigures its own reasoning process rather than directly producing conjectures or proofs:
– Switch from a "direct proof-first" mode to "counterexample-search-first" mode.
– Invoke external theorem provers to prune invalid proof paths.
– Deploy multiple agents specialized in algebraic manipulation vs. computational experiments.
Here, the search space is over *reasoning strategies themselves*.
**Search style:** Evolutionary or reinforcement-style optimization over proof-generation policies.

---

Figure 10: Illustration of meta-level ideation in mathematical reasoning.

## 4.4 Takeaways and Limitations

Inference-time scaling techniques expand the effective search space of a single LLM, giving it more opportunities to explore, refine, and evaluate potential ideas. Table 1 provides a comparison of scientific inference-time scaling methods. Empirical evidence from Monte Carlo Thought Search(Sprueill et al., 2023) demonstrates the impact of test-time scaling and structured search on scientific ideation quality. Using a domain-specific reward function, the study shows that increasing search depth and breadth substantially improves outcomes compared to single-pass prompting. While standard CoT achieves low rewards (2.04–2.27), adding self-consistency yields moderate gains (4.04–6.38). In

Table 1: Comparison of representative scientific inference time-scaling methods.

| Method | Search Mechanism | Feedback Source | Abstraction Level | Main Idea / Explanation |
|---|---|---|---|---|
| **Self-Refine** (Madaan et al., 2023) | Sequential refinement | Internal confidence | Hypothesis-level | Iterative self-critique loop refining outputs. |
| **PANEL** (Li et al., 2025d) | Sequential refinement | Internal confidence | Hypothesis-level | Stepwise reasoning refinement via critique generation. |
| **ResearchAgent** (Baek et al., 2024) | Sequential refinement | Peer agents | Hypothesis-level | Simulates multi-agent peer review for idea improvement. |
| **CriticAL** (Li et al., 2024b) | Sequential refinement | Internal confidence | Hypothesis-level | Applies statistical testing to critique and enhance reasoning. |
| **SCaSML** (Fan et al., 2025) | Sequential refinement | Scientific rules | Program-level | Incorporates physical laws; refines via Monte Carlo correction. |
| **Robin** (Ghareeb et al., 2025) | Sequential refinement | Simulator | Hypothesis-level | Refines ideas using simulated or in-lab feedback. |
| **IdeaSynth** (Pu et al., 2025) | Sequential refinement | Human + Tool | Hypothesis-level | Human-guided iterative idea refinement. |
| **Scideator** (Smith & Huang, 2025) | Sequential refinement | Human + Tool | Hypothesis-level | Joint human–tool refinement of idea components. |
| **AgentLab** (Schmidgall et al., 2025) | Sequential refinement | Peer agents | Hypothesis-level | Role-based pipeline from data analysis to paper drafting. |
| **Digital Twin** (Yang et al., 2025a) | Sequential refinement | Simulator + Peer agents | Hypothesis-level | Multi-agent tuning of simulation parameters. |
| **IRIS** (Feng et al., 2025) | Branching exploration | Peer agents + Human | Hypothesis-level | MCTS-based branching with both agent and human feedback. |
| **MC-NEST** (Rabby et al., 2024) | Branching exploration | Internal confidence | Hypothesis-level | Explores reasoning paths via MCTS with internal evaluation. |
| **MAGIC** (Xu et al., 2023a) | Branching exploration | Internal confidence | Hypothesis-level | Guides hypothesis branching via multi-armed bandit scores. |
| **MCTS** (Sprueill et al., 2023) | Branching exploration | Internal confidence | Hypothesis-level | Applies combinatorial MCTS for catalyst design. |
| **DGM** (Zhang et al., 2025a) | Branching exploration | Internal confidence | Program-level | Open-ended evolution: agents rewrite and test their own code. |
| **Casevo** (Jiang et al., 2024) | Sequential + Branching | Simulator + Peer agents | Hypothesis-level | Agent-based social simulation with adaptive opinion modeling. |
| **MOOSEChem2** (Yang et al., 2025b) | Parallel refinement | Internal confidence | Hypothesis-level | Parallel sampling of chemical hypotheses with internal scoring. |
| **ChemCrow** (Bran et al., 2023) | Programmatic planning | External tool | Program-level | Designs synthesis routes executed via cheminformatics APIs. |
| **CodeScientist** (Huang & Wu, 2024) | Programmatic planning | Simulator + In-lab run | Program-level | Writes lab scripts run in simulators or robotic labs. |
| **LLM-SR** (Shojaee et al., 2024) | Programmatic planning | External tool | Program-level | Generates and refines symbolic equations through tool feedback. |

contrast, structured search methods such as Tree-of-Thought (breadth-first search) achieve much higher rewards (9.91–13.8) by evaluating hundreds of candidate prompts (NP = 253, depth = 5). Monte Carlo Thought Search further improves performance (12.47–15.6) by deeper and more adaptive exploration (NP 301, depth 9). These results indicate that test-time scaling enables exploratory creativity by systematically traversing the ideation space, though gains come at the cost of increased computational expense and reliance on evaluator quality.

However, this search is ultimately confined to **one** model's internal distribution. For highly interdisciplinary or radically novel ideas, relying solely on a single model's representation may be insufficient. Just as breakthrough research often emerges when diverse human experts collaborate across fields, scientific ideation may require different LLMs—each trained or specialized in different domains—to interact, negotiate, and collectively build a richer space of ideas. While scientific discovery is not purely competitive, adversarial-style mechanisms—such as debate, critique, and counterexample generation—map naturally onto collaborative scientific workflows, where hypotheses are stress-tested, defended, and revised through peer interaction.

Thus, moving from single-agent scaling to multi-agent collaboration may bring LLM-assisted research closer to the collective, emergent creativity observed in real-world scientific communities (Chen et al., 2025b; Park et al., 2023). The next section explores these multi-agent architectures and their implications for scientific discovery.

# 5 Collaborative Multi-Agent Systems

As scientific inquiry becomes increasingly interdisciplinary, data-intensive, and empirically grounded, the limitations of a single LLM—no matter how capable—become more apparent. While previous strategies—such as prompt engineering, knowledge augmentation, and test-time scaling—enhance individual models, they do not fully address the inherent complexity of generating novel, valid, and feasible scientific hypotheses. This process often requires not just broad linguistic and conceptual knowledge, but also specialization, iterative reasoning, and interaction with real-world data or domain tools. Multi-agent LLM systems address this by simulating collaborative research environments, where multiple LLMs, each often imbued with specialized roles, debate, critique, and build upon each other's ideas.

From the perspective of complexity science, the behavior of such systems can exceed the sum of their parts: interactions among agents give rise to emergent properties not predictable from isolated individuals (Holland, 1998; Anderson, 1972). Extending this perspective to collaborative multi-agent LLMs, we can conjecture that debates among agents with diverse perspectives may not only expand the idea space but also challenge entrenched domain assumptions. In doing so, such systems could approach what Boden defines as *transformational* creativity Boden (2004)—that alters the very conceptual space in which ideas are generated—opening pathways to more radical and paradigm-shifting hypotheses. Although still speculative, this view resonates with the broader observation that emergent abilities tend to arise in sufficiently complex and interactive systems (Perumal et al., 2024; Chen et al., 2025b). Importantly, transitioning from single-agent to multi-agent systems (MAS) represents a shift in the *process* of creativity itself, introducing interactions and iterative reasoning as sources of innovation in the 4Ps framework (Rhodes, 1961).

There are two primary motivations for adopting multi-agent approaches in scientific discovery. The first one is **Automation of Complex Workflows**, where specialized agents divide and execute research tasks at scale. The second one is **Emergent Collaboration for Idea Generation**, where interacting agents critique and build upon each other's ideas to foster creativity and discovery. Together, these approaches represent a step toward building AI systems that are not only efficient assistants but also active, innovative collaborators in the scientific process.

## 5.1 Pipeline-Oriented Workflows: Automation of Complex Workflows

One prominent application of MAS is the creation of pipeline-oriented workflows, where specialized LLM agents collectively automate the various stages of scientific research. In these systems, each agent is assigned a distinct role—such as literature reviewer, hypothesis generator, experiment designer, or manuscript writer—mirroring the division of labor found in human research teams. By decomposing the scientific process into structured steps, these pipelines enable comprehensive coverage of the research lifecycle and improve scalability and efficiency.

Recent work has demonstrated the potential of this approach for fully autonomous discovery. *AI-Scientist* (Lu et al., 2024a) introduces an end-to-end framework that spans the entire process of scientific research: gathering and synthesizing literature, generating novel hypotheses, designing and simulating experiments, and even producing manuscripts and code repositories. *AI-Coscientist* (Gottweis et al., 2025), establishes a structured, multi-agent pipeline for autonomous scientific discovery. The system includes specialized agents for literature retrieval, idea generation, reflective self-evaluation, ranking and tournament-style comparison of hypotheses, and iterative refinement

through evolutionary search and debate. Similarly, *Robin* (Ghareeb et al., 2025) demonstrates a fully integrated chain of LLM agents that autonomously identify promising targets, design and run in silico experiments, and validate findings. This system notably proposed and computationally validated a novel therapeutic candidate for dry age-related macular degeneration, illustrating the real-world impact of multi-agent pipelines.

Infrastructure efforts such as *AgentLab* and *AgentRxiv* provide shared environments where autonomous agent laboratories can publish, retrieve, and iteratively build upon prior research results (Schmidgall et al., 2025; Schmidgall & Moor, 2025). These platforms introduce mechanisms for reproducibility and collaborative improvement, paralleling how human scientists interact through preprint servers and open-source repositories.

Pipeline-oriented MAS primarily target *efficiency* and *automation* rather than creativity or novelty. By systematically orchestrating specialized agents, they represent a significant step toward accelerating the entire research lifecycle and scaling scientific discovery to levels unattainable by human researchers alone. Future advances in model capabilities may reduce the need for explicit task division, but for now, these systems offer a practical path for handling the complexity and scale of modern scientific research.

### 5.2 Debate and Emergent Collaboration: Fostering Creative Discovery

While pipeline-oriented systems focus on efficiency and task automation, a different motivation for multi-agent approaches is to foster creativity and emergent idea generation. Scientific breakthroughs rarely arise from a single linear process—they are often the product of discussion, critique, and synthesis among researchers with different perspectives. LLMs, trained on vast corpora of human language, encode implicit models of human reasoning and social interaction. As shown in the *Generative Agents* framework (Park et al., 2023), when placed in narrowly defined contexts, LLMs can produce believable patterns of human-like behavior, from cooperation to competition. This suggests that a society of LLMs interacting through dialogue may exhibit emergent behaviors such as creativity, critical reasoning, and the collaborative problem-solving dynamics characteristic of real-world scientific communities (Piao et al., 2025; Piatti et al., 2024).

In this paradigm, agents do not simply pass outputs along a pipeline. Instead, they actively engage in multi-turn debates, critique one another's proposals, and iteratively refine ideas. Also, in collaborative MAS, additional LLMs are active participants in the creative process, so it stands in contrast to the single-agent generator-verifier setup (Section 4.2.1), where a verifier agent provides external feedback to guide the search space but remains a largely passive filter. This creates a richer search process, where hypotheses can be challenged, defended, and synthesized in multiple perspectives (Estornell & Liu, 2025).

A common architecture for such systems is the generator–discriminator loop, where one agent proposes ideas while others act as critics, rankers, or adversaries. Through repeated argumentation and revision, the group collectively converges on stronger and more coherent scientific hypotheses. For example, *VirSci* (Su et al., 2025) introduces a team of brainstorming agents, critics, and judges that iteratively debate to generate, evaluate, and refine hypotheses. By framing the process as structured rounds of argument and critique, this system demonstrates how collaboration between diverse roles can produce hypotheses with greater novelty and feasibility than single-agent approaches.

Similarly, the *Multi-Agent Debate (MAD)* framework (Liang et al., 2023) organizes agents into adversarial roles, where each proposal is rigorously scrutinized by competing agents and adjudicated by a neutral referee. This adversarial process is especially valuable for mitigating common LLM weaknesses such as hallucinations or factual inaccuracies, as challenges from other agents force the generator to strengthen its reasoning and evidence. The result is a kind of automated peer review, producing more resilient and reliable outputs.

Recent large-scale systems combine debate with structured pipelines. For instance, *AI-Coscientist* (Gottweis et al., 2025) incorporates generation agents, reflection agents, rankers, and evolutionary search to iteratively improve generated hypotheses. Here debate is deeply embedded within the pipeline itself: the generation agent engages in simulated scientific debates to iteratively refine and strengthen its proposals. Meanwhile, the ranking agent conducts tournament-style debate matches, where hypotheses compete head-to-head. In this way, debate serves a dual role—both as a filter to eliminate weak ideas and as a driver of exploration, expanding the search space toward diverse, high-quality hypotheses. This integration demonstrates how adversarial interactions can be harnessed within structured multi-agent pipelines, combining the efficiency of workflow automation with the creativity and rigor of collaborative scientific discourse.

This richer interaction structure allows the system to explore complex intellectual landscapes more fully and can help transcend purely combinatorial idea generation, enabling the emergence of exploratory and potentially groundbreaking hypotheses (Chen et al., 2025b).

### 5.3 Takeaways and Limitations

MAS have demonstrated strong potential for scientific discovery by providing two key benefits: scalable automation, where specialized agents divide complex workflows into coordinated stages (Su et al., 2025), and emergent creativity, where multi-turn debate and critique among agents generate novel exploratory hypotheses that resemble real-world scientific collaboration (Liang et al., 2023).

Empirical results from the VirSci study show that multi-agent systems consistently outperform single-agent baselines across both automated metrics and human evaluations, and this holds across different backbone models (GPT-4o, LLaMA-3.1-8B, and 70B). In particular, human ratings show clear gains in novelty (e.g., 3.78 vs. 3.10–3.22), feasibility (5.24 vs. 4.78–4.94), and effectiveness (4.95 vs. 4.43–4.77). At the same time, improvements in the proposed metrics (higher CI and lower CD; 7.1.1) indicate more coherent and less redundant ideation. Beyond benchmarked evaluations, a particularly notable result from the AI co-scientist (Gottweis et al., 2025) provides evidence of potential transformational creativity. The system independently recapitulated an unpublished gene-transfer mechanism in bacterial evolution via in silico discovery. As this mechanism was absent from public data and could not have been retrieved, the result goes beyond combinatorial recombination and points to the capacity of MAS to restructure the scientific search space, offering early empirical signals consistent with exploratory—and in rare cases, transformational—creativity.

However, MAS face notable limitations. Their reliance on brittle prompting and hand-tuned orchestration makes them fragile and susceptible to cascading failures (Cemri et al., 2025; tse Huang et al., 2025). Moreover, in line with *Sutton's Bitter Lesson* (Sutton, 2019), such human-engineered coordination mechanisms are often outperformed over time by simpler, more general approaches that scale with better-trained models and data. MAS also resemble other test-time scaling methods, relying on repeated inference and multi-round interaction to improve performance, which incurs

substantial computational overhead. As shown by Gao et al. (2025a), these advantages diminish as base LLMs improve, with strong single-agent models frequently matching MAS performance without communication costs. Consequently, MAS are best suited for settings where **interaction itself constitutes the primary source of value**, such as genuine multi-perspective debate or exploratory synthesis.

To move beyond experimental prototypes toward dependable systems, LLMs must become more capable, reliable, and efficient, reducing reliance on external coordination. This motivates a shift toward *parameter adaptation* methods that enhance models' internal reasoning abilities, enabling them to function as individually competent researchers without heavy orchestration or test-time compute overhead (Li et al., 2025c). The next section examines these approaches, focusing on how training-based techniques can improve hypothesis generation, reduce errors, and complement MAS by internalizing collaborative behaviors.

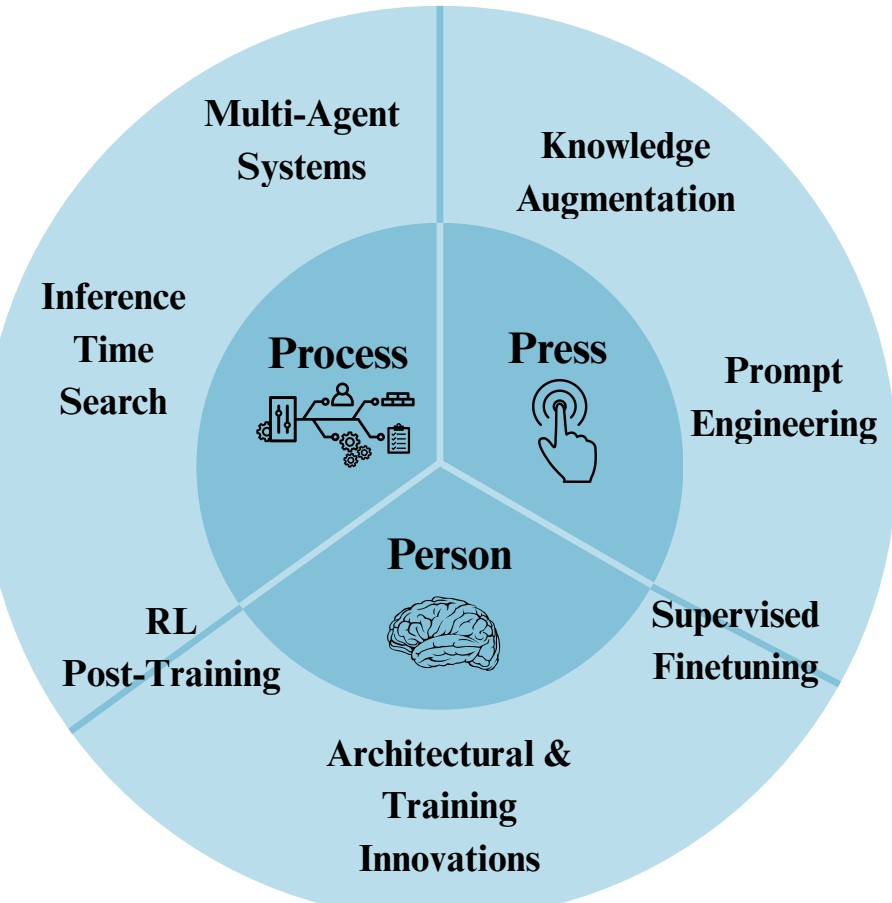

Figure 11: Conceptual mapping of LLM-driven scientific ideation methods to their primary source of creativity, inspired by Rhodes' 4Ps framework. Here, we focus on the generative dimensions—*person*, *process*, and *press*—as sources of creativity, with *product* reserved for evaluation. *Press* includes knowledge augmentation and prompt engineering, supplying external context and inspiration. *Process* covers multi-agent collaboration and inference-time search strategies that guide exploration. *Person* encompasses design choices and inductive biases in model architecture and training, including objectives, prediction formats, and alternative generative structures. Intersections illustrate methods that combine dimensions: supervised fine-tuning (person + press) internalizes external knowledge, while reinforcement learning (person + process) internalizes search strategies, enhancing the model's reasoning and creative capacities.

# 6 Parameter-Level Adaptations

As scientific AI moves from experimental prototypes toward dependable infrastructure, underlying LLMs must become smarter, more reliable, and more efficient. In such settings, reliance on test-time scaling or external tool use—including multi-agent interactions—becomes increasingly impractical, as these approaches incur high computational costs, latency, and fragility. To overcome these limitations, it is necessary to internalize both domain knowledge and effective creative pathways within the model itself. This represents a paradigm shift from relying on *process* as the main source of creativity to cultivating the *person* itself, as conceptualized in the 4Ps framework (Rhodes, 1961). By incorporating both knowledge and search capabilities into the model, these approaches amortize the cost of exploration: once trained, the model can generate high-quality hypotheses without repeated costly inference-time interactions (Li et al., 2025c).

In this section, we systematically categorize parameter adaptation methods into three groups: **Supervised Fine-Tuning Approaches**, which leverage curated datasets to teach domain-specific knowledge; **Reinforcement Learning-Based Methods**, which optimize generative and search behavior through feedback; and **Hybrid and Preference Optimization Methods**, which combine supervised and reinforcement signals to align models with both task objectives and creative evaluation criteria.

## 6.1 Supervised Fine-Tuning Approaches

Supervised fine-tuning (SFT) represents one of the earliest and most widely adopted strategies for adapting LLMs to specific downstream tasks. In the context of scientific discovery, SFT leverages labeled datasets—often curated from scientific literature, domain-specific corpora, or structured instruction-following examples—to imbue models with domain expertise and task-relevant capabilities. The strength of SFT lies in its ability to exploit existing pretraining knowledge and refine it for specific scientific domains, such as chemistry, biology, or physics. However, SFT can also face limitations, including overfitting to narrow datasets or failing to generalize to novel problems outside the training distribution. Additionally, curating high-quality, large-scale datasets remains one of the most challenging aspects of using SFT, as it is both costly and labor-intensive. Recent advances have sought to mitigate these challenges by introducing techniques such as instruction generation, curriculum learning, and fine-grained data curation. As a result, SFT continues to be a cornerstone of training pipelines for scientific LLMs, particularly when paired with robust evaluation and generalization strategies.

A notable direction within SFT is **Domain-specific Adaptation**. The *DARWIN* (Xie et al., 2023) series leverages SFT on scientific literature and structured scientific datasets through an automated Scientific Instruction Generation (SIG) model. By applying SFT on over 20,000 instruction–answer pairs, the model is aligned with ground-truth knowledge in scientific domains, leading to substantial performance improvements across tasks in physics, chemistry, and materials science. Building on this paradigm, *OmniScience* (Prabhakar et al., 2025) introduces a more comprehensive framework for domain specialization through (1) domain-adaptive pretraining on curated scientific corpora, (2) instruction tuning with tailored datasets for domain-specific tasks, and (3) reasoning-based knowledge distillation to strengthen logical consistency and contextual relevance. Scenario 12 illustrates this by showing how curated datapoints from OmniScience and DARWIN help models shift from inconsistent answers to domain-consistent scientific reasoning.

---

**Scenario:  SFT-Enhanced Question Answering in Materials Science**

A base LLM is asked: *"When photons with energies just below* $2.3\,eV$ *interact with a thin film whose absorption sharply increases above this threshold, what happens?"*
Without finetuning, the model often mispredicts (e.g., "reflection" or "photoluminescence").
After SFT, the model succeeds — because its training set included domain-grounded datapoints such as:

---

**OmniScience datapoint**

**Instruction (input):** Read the following excerpt from a materials science paper and answer the MCQs about its optical properties:
*Excerpt:* "The absorption coefficient of the thin film increases sharply around 2.3 eV, indicating the onset of strong interband transitions between the valence and conduction bands. Below this energy, the film shows very low absorption, which suggests that it behaves transparently in the visible region."
*Question:* What happens when photons with energies just below 2.3 eV interact with the film?
A) They are strongly absorbed via interband transitions.
B) They are mostly transmitted (i.e. not absorbed).
C) They cause photoluminescence.
D) They are reflected with a high reflection coefficient.

**Response (label):** B) They are mostly transmitted (i.e. not absorbed).
**Reasoning:** Below $\sim 2.3\,eV$, the absorption coefficient is very low, which means the film does not absorb much; hence photons in that energy range are largely transmitted.

---

**DARWIN datapoint**

**Instruction (input):** Given a description of a material, classify whether it behaves as a conductor, semiconductor, or insulator.
**Material:** Silicon at room temperature.

**Response (label):** Semiconductor.
**Reasoning:** At room temperature, silicon has a bandgap of about 1.1 eV, which is small enough to allow limited electron excitation into the conduction band, characteristic of a semiconductor.

---

Figure 12: SFT improves LLM understanding of physical principles in materials science question answering tasks.

Extending this approach to chemistry, *ChemLM* (Kallergis et al., 2025) applies domain-adaptive pretraining followed by SFT on chemical datasets, enabling accurate molecular property prediction and identification of potent pathoblockers. Unlike conventional SFT, ChemLM leverages chemical SMILES sequences as input, treating molecules as a language to capture structural patterns and relationships, which supports discovery of novel compounds. *ChemMLLM* (Tan et al., 2025) further enhances this paradigm through multimodal SFT across text, SMILES, and molecular images. By integrating these modalities, ChemMLLM achieves unified reasoning over chemical structure, function, and visual representation, outperforming GPT-4o by 118.9% on image-based molecule optimization tasks. These efforts demonstrate that domain-specific SFT, when combined with modality-aware pretraining, can substantially accelerate discovery and reasoning while maintaining high accuracy.

## 6.2 Reinforcement Learning-Based Methods

Reinforcement learning (RL) has emerged as a pivotal approach for aligning LLMs with complex, goal-driven tasks in scientific discovery. In traditional reinforcement learning, a *Markov Decision Process (MDP)* serves as the mathematical framework for representing the key components of an RL algorithm: states, actions, transition probabilities, rewards, and the discount factor. In the context of language models' post-training, these elements are analogous to the input prompt plus the tokens generated so far (states), selecting the next token (action), transition probabilities conditioned on state and action , and the accuracy of the generated response for a given query (reward). RL has been widely adopted to enhance the reasoning capabilities of LLMs within this framework (DeepSeek-AI et al., 2025)(Wang et al., 2023b)(Shao et al., 2024). Traditional RL frameworks have primarily focused on optimizing terminal rewards, such as achieving the correct solution or maximizing accuracy. However, scientific discovery and hypothesis generation are inherently process-oriented: intermediate steps such as problem decomposition, hypothesizing alternatives, self-correction, experiment design for hypothesis validation and citing credible sources are just as critical as the final output. To address this, an increasing body of research is shifting toward *PRMs* (Lightman et al., 2023), which evaluate not only "what" models say but also "how" they think. *ORMs*, though easier to implement, tend to favor shallow heuristics and are susceptible to reward hacking. In contrast, PRMs provide granular, intermediate feedback that fosters deeper chains of reasoning. This evolution in RL design—from black-box evaluation to interpretable, step-wise reward shaping—enables new paradigms like debate, self-play, and simulation-based learning, all of which are increasingly relevant for equipping LLMs with the capacities for rigorous and transparent scientific reasoning.

### 6.2.1 Algorithm Discovery and Reasoning

RL has been instrumental in enabling LLMs to discover efficient algorithms and enhance scientific reasoning capabilities. For instance, Surina et al. (2025) demonstrated that RL-based fine-tuning, integrated within evolutionary search processes, dynamically improves LLM performance in algorithm discovery tasks. By using *FunSearch* (Romera-Paredes et al., 2024) (only evolutionary search without RL) as a baseline, they showed that the combination of RL finetuning with evolutionary search can enhance the performance of the model and surpass the baseline in constructing optimal algorithms for tasks like bin packing, traveling salesman problem and flatpack. Additionally, leveraging expert domain knowledge, RL-based fine-tuning has been successfully applied to genomics, where expert forum discussions are automatically transformed into RL-friendly multiple-

Table 2: Parameter-adaptation methods for scientific idea generation and discovery.

| Method | Field | Adaptation strategy | Main contribution / result | Main challenges |
|---|---|---|---|---|
| **Cycle-Researcher** (Weng et al., 2025a) | Automated Scientific research | Iterative preference training (SimPO-based RL between policy and reward) | • Automates hypothesis–review cycles
• Produces ideas near accepted paper quality | • Limited cross-domain generalization
• Reward instability |
| **Dynamic Control** (Li et al., 2024c) | Scientific idea generation | SFT → PPO-based controllable RL with multi-objective rewards (novelty, feasibility, usefulness) | • Dynamically adjusts idea generation
• Balances novelty and feasibility | • Reward hacking risk
• Trade-offs between objectives |
| **CRPO** (Ismayilzada et al., 2025a) | General creative generation | Preference-based RL (modular DPO) using creativity metrics (novelty, diversity, surprise, quality) on MUCE dataset | • Boosts originality and diversity with quality retention
• Outperforms DPO, DDPO, GPT-4o | • LLM-judge vulnerability
• Static preference bias |
| **Algorithm Discovery** (Surina et al., 2025) | Computational scientific discovery | Hybrid evolutionary search + RL fine-tuning (DPO + forward KL) | • Discovers novel algorithms for TSP, Bin Packing, regression
• Improves optimality and diversity | • High computation cost
• Large search space
• Limited interpretability |
| **DARWIN Series** (Xie et al., 2023) | Scientific reasoning | Domain SFT on ~60K tasks → multi-task fine-tuning with synthetic instruction generation | • Improves domain reasoning and factual acc
• Strong on SciQ benchmark | • Domain overfitting
• Spurious learning
• Data diversity dependence |
| **Smiley-Llama** (Cavanagh et al., 2025) | Domain-specific discovery, molecular | SFT → DPO → RL (iMiner) for property-optimized molecule design | • Generates valid, diverse molecules
• Higher novelty and diversity than RNN baselines | • Dataset bias, limited coverage
• Costly validation |
| **Chem-LM** (Kallergis et al., 2025) | Domain-specific discovery, chemistry | Self-supervised molecular pretraining → domain adaptation → SFT for property prediction | • Improves molecular property prediction
• Aids discovery of chemical candidates | • Limited interpretability
• Poor cross-family generalization |
| **Chem-MLLM** (Tan et al., 2025) | Domain-specific discovery, chemistry | Multimodal fine-tuning on text, SMILES, and molecular images; unified encoder–decoder design | • Bridges modalities for visual–text–structure reasoning
• Enables molecule generation/ mapping | • Complex alignment
• High compute demand
• Pretraining bias |

choice questions (MCQs), significantly enhancing the model's scientific reasoning abilities (Yin et al., 2025b).

### 6.2.2 Domain-Specific Reasoning and Optimization

RL has also been successfully adapted for domain-specific scientific challenges that demand interpretability or structured optimization.

In protein discovery, RL has been coupled with *Protein Language Models (pLMs)* to design novel biologically plausible sequences optimized for therapeutic objectives. For instance, Stocco et al. (2025) demonstrate how PPO-based fine-tuning can produce protein sequences with improved binding affinity and stability, including candidates for EGFR binders. Similarly, Subramanian et al. (2024) applies policy-gradient training on top of pre-trained protein models to generate diverse sequences tailored to objectives such as structural integrity. These studies illustrate how combining RL with pLMs enables protein discovery beyond natural sequence space.

In drug discovery, RL leverages structured and composite rewards to optimize molecular candidates. DrugImproverGPT introduces *Structured Policy Optimization (SPO)* to align molecules with therapeutic objectives while preserving beneficial properties (Liu et al., 2025d). Similarly, PPO-based fine-tuning balances protein-target efficacy and chemical validity, enhancing novelty and drug-likeness (Ahmed & Mohammed, 2025).

### 6.3 Hybrid and Preference Optimization Methods

Hybrid and preference optimization approaches integrate SFT, RL, and advanced alignment methods to balance complex, multi-dimensional objectives in scientific discovery. These frameworks combine the grounding benefits of SFT with the adaptive refinement capabilities of RL, while also leveraging preference-based optimization to address challenges of diversity, creativity, and precision.

One prominent application is research idea generation, where a two-stage framework combines SFT with controllable RL to dynamically balance novelty, feasibility, and effectiveness through multi-dimensional reward modeling (Li et al., 2024c). Similarly, in molecular generation, a two-stage pipeline of SFT followed by RL significantly enhances structural diversity in generated molecule sets, outperforming conventional decoding methods (Jang et al., 2025).

Preference optimization has emerged as a particularly effective complement to these hybrid strategies. *DPO* (Rafailov et al., 2023) simplifies alignment by directly learning from human preference pairs, avoiding the instability of complex reward models. *SmileyLlama* (Cavanagh et al., 2025), derived from the Llama-3.1-8B-Instruct model, employs a pipeline of SFT combined with DPO to generate drug-like molecules optimized for biological activity and validated efficacy against protein targets. However, standard DPO often favors "safe" high-reward responses, limiting creativity and diversity in outputs.

To address this, several recent works propose creativity- and diversity-enhanced preference optimization. *CrPO* introduces a multi-dimensional alignment framework that injects signals of novelty, diversity, surprise, and quality into the optimization process, producing outputs that are not only accurate but also more surprising and original (Ismayilzada et al., 2025a). *DivPO* extends this idea by explicitly selecting preference pairs to reward rare but high-quality responses while penalizing common low-quality ones, leading to substantial improvements in output diversity without sacri-

ficing quality (Lanchantin et al., 2025). These approaches are particularly relevant for scientific idea generation, where generating multiple distinct but plausible hypotheses is as important as generating feasible ones.

A recent example of this direction is *CycleResearcher*, which proposes a complete framework for automating the research cycle with open-source LLMs (Weng et al., 2025a). The framework integrates two models: CycleResearcher, responsible for literature review, idea development, manuscript generation, and refinement; and CycleReviewer, which simulates peer review and provides iterative feedback via RL. The models are trained on two new datasets—*Review-5k*, containing structured peer reviews, and *Research-14k*, which provides outlines and segmented academic papers for fine-tuning. Through an iterative preference optimization approach (SimPO), the CycleResearcher–CycleReviewer loop allows the policy model to improve over multiple iterations, with CycleReviewer acting as a reward model to guide preference-based updates. Unlike standard DPO, which tends to reward only "safe" responses, this iterative design promotes more nuanced alignment with academic standards. In practice, the framework achieved competitive review scores, showing that LLMs can produce outputs near the quality of human preprints while still leaving room for further improvement.

Structured optimization methods further extend this landscape by employing more specialized alignment techniques. *Energy Rank Alignment (ERA)* (Chennakesavalu et al., 2025) uses gradient-based optimization to align molecular and protein generation with explicit chemical objectives, balancing robustness, diversity, and property alignment. *Contrastive Preference Optimization (CPO)* (Gkoumas, 2024) improves precision by explicitly contrasting optimal and suboptimal outputs, thereby enhancing reliability and mitigating issues like memorization or hallucination.

In summary, the structured exploration of training-based methods—including RL, SFT, hybrid approaches, structured optimization, and efficiency-focused curriculum learning—highlights the rich landscape of opportunities for adapting LLMs for scientific discovery. These methodologies collectively enable precise alignment of model capabilities with specific scientific discovery objectives, thereby significantly advancing automated scientific research. A concise overview of the most recent parameter-adaptation frameworks that target scientific-idea generation or discovery is provided in Table 2.

## 6.4   Takeaways and Limitations

Post-training adaptation of LLMs—using SFT, RL, or hybrid/preference-optimization methods—is a powerful tool for scientific idea generation, but each approach has distinct limitations.

**SFT** instills reliable reasoning patterns and aligns outputs with human demonstrations, ensuring consistency and interpretable results on tasks close to the training distribution. However, it is constrained by **static datasets** and can **overfit** to dataset biases, limiting generalization, handling of ambiguous prompts, and exploration of novel or high-risk hypotheses (Chu et al., 2025; Hua et al., 2025a).

**RL** guides exploration toward task-specific objectives, enabling creativity and risk-taking for potentially groundbreaking hypotheses. Its challenges include *reward over-optimization* (model focuses on easily achievable rewards, limiting novelty), *reward hacking* (model exploits loopholes in the reward function to get high reward without truly solving the task), *output instability* (early training can produce incoherent or repetitive outputs), *sensitivity to reward design* (poorly specified

rewards may penalize creative or cross-domain ideas), and *computational cost* (multiple sampling and updates make large-scale training expensive) for large-scale discovery (Mohammadi, 2024; Kirk et al., 2023; DeepSeek-AI et al., 2025).

**Hybrid/Preference-Based Methods** (e.g., DPO, CRPO) balance reliability and creativity, improving quality and multi-objective control. Yet, they face *system complexity* (multi-stage pipelines are harder to debug, tune, and maintain), *bias propagation* (preferences from human- or AI-generated data can introduce systemic biases), *cross-domain generalization* (models optimized on one domain may underperform on others), and *feedback sensitivity* (quality, consistency, and granularity of preference signals directly affect creativity and exploration of unconventional ideas). that can constrain exploration of unconventional ideas.

Empirical evidence supports this characterization. In the CycleResearcher (Weng et al., 2025a) study, parameter-adapted models consistently outperform the AI Scientist baseline in peer-review–style evaluation, achieving higher overall scores (5.15–5.38 vs. 4.31) and non-zero acceptance rates (24–35% vs. 0%). At a criterion level, adapted models show improvements in *soundness*, *presentation*, and *contribution*, with the latter—interpreted as a proxy for novelty—improving from 2.15 to 2.53–2.60 approaching the performance of human-reviewed preprints across all three dimensions. Complementary evidence from the CrPO (Ismayilzada et al., 2025a) study, evaluated using NoveltyBench (Zhang et al., 2025c), further clarifies while SFT substantially improves quality-interpreted as valueness- (5.0 vs. 3.1–4.1 for base LLMs vs. 4.3–4.9 for CrPO) but yields more limited gains in novelty (5.7 vs. 4.0 for base LLMs), remaining well below preference based approaches (e.g., CrPO: 6.8–7.9). These results indicate that parameter adaptation primarily strengthens valueness, with stronger novelty emerging when optimization explicitly rewards exploration. Further comparisons can be found in Table 3.

In summary, SFT ensures consistency, RL drives exploration, and hybrid methods balance both. While these approaches face challenges such as dataset biases, reward mis-specification, overfitting, and limited generalization, they **internalize** creativity and domain knowledge within the model itself. This reduces reliance on test-time computation or external resources and shifts the source of creativity from purely *Process* or *Press* signals toward the model's internal capabilities, aligning with the *Person* aspect of 4p framework (Rhodes, 1961). Addressing the limitations while leveraging these intrinsic strengths could be the key to unlocking LLMs' full potential for scientific hypothesis generation and discovery.

Table 3: Method categories and their expected effects on novelty and valueness in scientific ideation. Expected effects reflect empirically observed trends and theoretically motivated interpretations reported in prior work, rather than guaranteed outcomes.

| Method Category | Primary Mechanism (4Ps) | Expected Effect on Novelty | Expected Effect on Valueness | Representative Works |
|---|---|---|---|---|
| Knowledge & Retrieval Augmentation | External knowledge injection (Press) | ↑/↔ Combinatorial novelty via recombination of retrieved concepts; may overfit to retrieved context | ↑ Grounding, correctness, and feasibility; reduced hallucinations | Section 2.3 (Gu et al., 2025; Gao et al., 2025b) |
| Prompt-based Distributional Steering | Input-level guidance (Press / Process) | ↑ Novelty by steering generation toward lower-probability regions (constraints, role-play, reframing) | ↑/↓ Structured prompting can improve reasoning; misaligned constraints may degrade value | Section 3.5 (Zhao et al., 2025; Li et al., 2024a) |
| Inference-time Scaling & Search | Systematic exploration beyond single-pass decoding (Process) | ↑↑ Exploratory novelty via branching, refinement, and structured search | ↑/↓ Improves with evaluation or pruning; unconstrained search risks invalid ideas | Section 4.4 (Sprueill et al., 2023; Feng et al., 2025) |
| Multi-agent & Deliberative Systems | Interaction, debate, and critique (Process; proxy for Person) | ↑↑↑ Potential for transformational novelty via emergent interactions | ↑ Improved coherence, plausibility and even robustness through iterative critique | Section 5.3 (Gottweis et al., 2025; Johnson & Davis, 2024) |
| Parameter Adaptation & Learning | Internalization of knowledge and strategies (Person) | ↔/↑ SFT, which reinforces training patterns; RL or PO via internalized exploration | ↑ Consistency, feasibility, and robustness through internalized constraints | Section 6.4 (Weng et al., 2025a; Ismayilzada et al., 2025a) |

## 7 Evaluation of Scientific Idea Generation: Frameworks and Challenges

Evaluating LLM-generated scientific ideas is crucial yet difficult. Although evaluation mechanisms share techniques and sources with the feedback systems discussed in Section 4.2, they serve different purposes: feedback guides idea generation through search and refinement, while evaluation assesses final outputs to determine a model's effectiveness as a scientific ideator. In Rhodes' 4Ps (Rhodes, 1961) framework, the *Product* dimension—novel, useful, and correct outputs—serves as the key endpoint of creativity, but assessing it is challenging because scientific ideas are open-ended and their true impact unfolds over time. This makes evaluation a major bottleneck: expert judgment is insightful but unscalable, automated metrics are efficient but reductive, and LLM-as-a-judge

approaches promise a balance while still inheriting biases. In this section, we survey three major families of evaluation frameworks: (1) **Computational and Execution-based** metrics, which operationalize dimensions of creativity such as novelty, diversity, feasibility and utility into quantifiable scores; (2) **Expert review**, which captures nuanced human intuition beyond what metrics alone can measure; and (3) **LLM-as-a-judge** models, which aim to synthesize the advantages of automation and human judgment. Table 4 summarizes *Evaluation paradigms*, highlighting their core strategies, strengths, and limitations.

## 7.1 Computational and Execution-Based Metrics

Computational and Execution-Based Metrics translate creativity dimensions—novelty, valueness (diversity, feasibility, and impact)—into measurable, reproducible signals; however, we emphasize that these metrics provide indirect proxies for creativity rather than direct measurements of creative insight, and should be interpreted as operational indicators rather than exhaustive characterizations. Computational metrics derive these signals analytically through **Model-based formulations**, while execution-based metrics validate them empirically through **Simulation or real-world testing**. Early work, such as *SciMon* (Wang et al., 2024a), evaluates the overall quality of generated ideas using surface-level text similarity measures (e.g., rouge, bertscore, bartscore), without distinguishing between different aspects of creativity to compare generated ideas against reference texts. While such holistic NLP-based evaluations offer scalability and ease of automation, they fail to capture the conceptual depth and open-ended multidimensional nature of scientific creativity.

Recent research instead decomposes creativity into distinct dimensions, each requiring specialized evaluation strategies: for instance, novelty and diversity are better assessed through *semantic distance* or representational density, whereas feasibility and impact are more meaningfully estimated through *execution-based or experimentally grounded* validation.

However, while computational and execution-based metrics enable systematic, high-throughput evaluation, they provide only a partial view of creativity and often overlook the contextual insight and *intuitive judgment* that experts contribute. Together, they form a more fine-grained framework for assessing scientific idea generation, which we review in detail below.

### 7.1.1 Novelty and Originality

One of the most widely studied criteria in evaluating the scientific creativity of LLMs is *novelty*, often operationalized as the degree to which an idea departs from existing knowledge or occupies a sparse region of the scientific landscape. Recent works have approached novelty measurement through a variety of perspectives, including *semantic density* and *knowledge graph triplets*.

Several recent studies have approached novelty assessment through semantic local density, assuming that truly novel ideas should lie in sparse regions of the scientific embedding space. Su et al. (2024) introduced two such measures: **Historical Dissimilarity (HD)**—the average Euclidean distance between a generated idea's embedding and its five most similar historical abstracts—and **Overall Novelty (ON)**, which further incorporates the contemporaneity and impact of neighboring works:

$$\text{ON} = \frac{\text{HD} \times \text{CI}}{\text{CD}}, \tag{1}$$

where CI denotes the average citation count of the five most similar contemporary papers and CD their semantic dissimilarity. But, such density-based scores are sensitive to domain-specific

Table 4: Comparison of evaluation paradigms for scientific idea generation.

| Evaluation paradigm | Representative methods | Example works | Strengths / limitations | Creativity Dimension |
|---|---|---|---|---|
| **Human Judgment** | • Expert panel, real peer-review submission 
 • Consensual Assessment Technique 
 • Structured Likert-scale | **Scideator** (Smith & Huang, 2025), **Nova** (Hu et al., 2024), **AI Scientist v2** (Yamada et al., 2025), **CAT** (Amabile, 1982), **AIdeation** (Wang et al., 2025d) | + Conceptual depth, captures nuanced aspects 
 - Time-consuming, costly, subjective, unscalable | Feasibility, Impact |
| **LLM-as-a-Judge** | • Prompt-based judges 
 • Fine-tuned judges 
 • Reinforcement-optimized reward models 
 • Hybrid frameworks | **LiveIdeaBench** (Ruan et al., 2024), **CycleReviewer** (Weng et al., 2025a), **DeepReviewer** (Zhu et al., 2025b) **HARPA** (Vasu et al., 2025), **ReviewRL** (Zeng et al., 2025) | + Scalable, reproducible, approximates expert reasoning and intuition 
 - Sensitive to prompts, training bias, and domain drift | Feasibility, Novelty, Impact |
| **Computation Based** | • Text similarity metrics: BLEU, BERTScore, DSI 
 • Semantic distance: RND, HD, ON, APD, Vendi Score 
 • Graph centrality 
 • Bayesian fusion | **SciMon** (Wang et al., 2024a), **CrPO** (Ismayilzada et al., 2025a), **VirSci** (Su et al., 2024), **RND** (Wang et al., 2025e), **Vendi Score** (Friedman & Dieng, 2023), **SciND** (Gupta et al., 2024; Ke et al., 2023), **HypoAgent** (Duan et al., 2025) | + Objective, interpretable, domain-agnostic, scalable 
 - Surface-level semantics; depends on corpus quality and embedding fidelity | Novelty, Diversity, Impact |
| **Execution Based** | • Simulators, compilers, autonomous labs 
 • Behavioral execution | **AI Scientist** (Lu et al., 2024a), **AI Co-Scientist** (Gottweis et al., 2025), **MindShift** (Wu et al., 2024), **LLM Counter Speech** (Podolak et al., 2024) | + Grounded, outcome-based validation 
 - Expensive, domain-limited, low scalability | Feasibility, Impact |

corpus variance—ideas in dense fields (e.g., NLP) appear less novel than those in sparse ones (e.g., quantum biology).

Building on this intuition, **Relative Neighbor Density (RND)** evaluates novelty through the density of an idea's semantic neighborhood relative to its neighbors' own densities (Wang et al., 2025e). Given a set of ideas $I = \text{idea}_i$, each represented by an embedding $v_i = G(\text{idea}_i)$ obtained via a semantic encoder $G(\cdot)$, RND computes for each idea $i$ its $P$ nearest neighbors in the literature corpus $A$:

$$v_1, v_2, \ldots, v_P = \text{KNN}(v_i, P, A), \tag{2}$$

and estimates the local neighbor density:

$$\text{ND}(v_i) = \frac{1}{Q} \sum_{k=1}^{Q} d(v_i, v_k),$$

where $d(\cdot, \cdot)$ is the cosine distance.

Novelty of $v$ is quantified by comparing the candidate's ND with the average ND of its neighbors:

$$\text{RND}(v_i) = \frac{\frac{1}{|N(v_i)|} \sum_{y \in N(v_i)} \text{ND}(y)}{\text{ND}(v_i) + \varepsilon},$$

where $N(v)$ is the neighborhood set and $\varepsilon$ is a small constant for numerical stability. A higher score indicates that the candidate lies in a sparser region relative to its peers. For instance, if an abstract has ND = 0.12 while its neighbors average ND = 0.36, the ratio RND = 3.0 suggests strong novelty. Unlike HD and ON, which measure novelty in absolute terms, RND provides a *relative* normalization—mitigating cross-domain bias by comparing each idea's isolation to that of its immediate semantic neighbors. Empirical validation on large-scale datasets (PubMed and arXiv) showed that RND achieves AUROC > 0.80 across computer science and biomedical datasets, demonstrating robustness.

An alternative symbolic perspective is adopted in the **SciND** benchmark (Gupta et al., 2024), which casts novelty detection as the identification of new knowledge-graph triplets. Here, scientific findings are represented as (subject, relation, object) triplets extracted from the literature. A triplet is labeled novel if it does not appear in the historical scientific knowledge graph. For example, ("transformer architecture", "uses", "sparse attention routing") would be marked novel if prior graphs contained only ("transformer architecture", "uses", "softmax attention"). This approach offers interpretable, fine-grained novelty signals at the entity-relation level, though it depends critically on reliable relation extraction and graph coverage.

Taken together, these methods capture complementary aspects of novelty: ON measures absolute semantic distance, RND provides relative local density, and SciND adds symbolic, conceptually grounded interpretability. Combining these signals enables robust, multidimensional evaluation of originality in LLM-generated scientific ideas.

### 7.1.2 Diversity

While novelty measures how far a single idea departs from prior knowledge, *diversity* evaluates the breadth of a set of generated ideas. A model that outputs only a narrow cluster of novel but similar

hypotheses may *underexplore* the solution space. Diversity thus ensures that systems span multiple conceptual directions, not just isolated rare points. Empirical studies confirm that high novelty does not imply diversity—models may generate unique yet domain-concentrated ideas—underscoring the need to evaluate both dimensions jointly (Ruan et al., 2024).

A common strategy for measuring diversity is to compute the **Average Pairwise Dissimilarity (APD)** among generated responses in a shared embedding space. Let $Y = \{y_1, \ldots, y_M\}$ denote a set of $M$ generated ideas, each represented by embedding $v_i$. A straightforward diversity score is

$$\text{Div}(Y) = \frac{2}{M(M-1)} \sum_{i<j} \big(1 - \cos(v_i, v_j)\big),$$

Larger values indicate that the responses are semantically spread out. For example, if five candidate hypotheses include one in physics, one in computational biology, and three variations of the same NLP idea, the mean pairwise similarity would be skewed upward by the NLP cluster, leading to a lower overall diversity score.

Building on the same principle, instead of evaluating the whole set at once, *CrPO* Ismayilzada et al. (2025a), computes a per-response dispersion score. Responses with higher score—i.e., those further from their peers—receive stronger rewards, encouraging models to cover multiple directions rather than collapsing to a single theme.

Beyond cosine-based measures, **Vendi Score** Friedman & Dieng (2023) provides a more principled measure of set distinctness using Shannon entropy over the eigenvalues of the similarity matrix. Given a similarity matrix $K \in \mathbb{R}^{M \times M}$ constructed from embeddings, let $\{\lambda_1, \ldots, \lambda_M\}$ denote its normalized eigenvalues (summing to 1). The Vendi Score is defined as:

$$\text{Vendi}(Y) = \exp\bigg( -\sum_{i=1}^{M} \lambda_i \log \lambda_i \bigg).$$

This measure can be interpreted as the "effective number of distinct items" in the set. For example, if three ideas are nearly identical, their embeddings produce a rank-1 similarity matrix, yielding $\text{Vendi}(Y) \approx 1$, despite $M = 3$. Conversely, if the three ideas are mutually orthogonal, the eigenvalues are uniform and $\text{Vendi}(Y) = 3$. This property makes the Vendi Score robust to redundancy.

In summary, APD provides a simple and scalable measure of set-level spread, CrPO computes a per-response dispersion score and Vendi Score adds redundancy sensitivity. Together, these approaches ensure that LLMs explore the scientific idea space broadly rather than converging prematurely on narrow regions.

### 7.1.3 Feasibility and Resource-Efficiency

While novelty and diversity measure idea exploration, *feasibility* and *resource-efficiency* concern execution—whether a generated idea can be realistically implemented given scientific or resource constraints. In some works, feasibility is indirectly estimated from prior knowledge by comparing new ideas to historically validated ones. These retrieval-based approaches rely on LLM critics to infer plausibility, as discussed in Section 7.3.1. However, a growing class of methods assess feasibility either computationally—by statistical models—or more directly by executing or simulating them in real or virtual environments, providing grounded, outcome-based validation.

The *HypoAgents* framework (Duan et al., 2025) proposes an **N–R–F** (Novelty–Relevance–Feasibility) model that quantifies idea quality via Bayesian inference and information fusion. Using retrieved literature and semantic embeddings, it measures feasibility as the Bayesian probability of supporting evidence—deriving scores *computationally.*

*Execution-grounded* evaluations appear most clearly in autonomous research pipelines. For instance, *AI Scientist* papers (Lu et al., 2024a; Yamada et al., 2025) or *DeepScientist* (Weng et al., 2025b) integrates idea generation, code synthesis, and experimental validation in a closed loop. Hypotheses proposed by one agent are automatically implemented as machine-learning experiments, trained, and benchmarked; success metrics such as performance gain or reproducibility indicate feasibility. Similarly, *HARPA* (Vasu et al., 2025) introduces a testability-driven ideation framework, where an LLM reward model predicts feasibility scores and selected hypotheses are subsequently verified through simulation or real-world trials, closing the loop between idea generation and empirical testing.

In materials and molecular science, execution-based feasibility is operationalized through simulation and experimental pipelines. Borg et al. (2023) introduce empirical measures such as **Discovery Yield**—the fraction of target materials discovered—and **Discovery Probability**—the likelihood of identifying at least one viable candidate—both derived from iterative experimental cycles. Similarly, Gallagher & Webb (2025) quantify efficiency as the proportion of candidate compounds passing synthesizability, stability, or toxicity constraints during virtual screening, effectively measuring the fraction of computationally or experimentally "non-wasted" recommendations. Systems like *BenchML* (Poelking et al., 2021) combine simulators and synthesis predictors to test chemical hypotheses, evaluating feasibility based on success rates such as molecular stability or docking performance.

Overall, these execution-grounded approaches provide the most direct signal of feasibility and resource-efficiency—measuring not whether an idea "sounds plausible", but whether it "works". However, execution-based validation is limited to domains with accessible simulators or experimental infrastructure, and many scientific fields lack fast or general-purpose implementation platforms. Consequently, scalable evaluation frameworks often combine them with llm-as-a-judge (Section 7.3), balancing empirical rigor with throughput.

### 7.1.4 Utility and Impact

Whereas feasibility measures whether an idea is plausible and resource-efficient evaluation captures its cost-effectiveness, *impact* aims to assess the *significance* of a scientific idea once realized. An idea may be technically feasible yet trivial in its contribution, or conversely, difficult to achieve but transformative in its outcomes. Impact evaluation is particularly challenging because it often requires long-term or context-specific signals, yet several frameworks have emerged to provide measurable proxies.

A computational line of work estimates the impact through **Bibliometric and Scientometric** indicators, treating citation counts, $h$-index contributions, and network centrality scores as a proxy for influence. Formally, given a knowledge graph $G = (V, E)$ of previous publications, the impact of a new idea $y$ can be approximated by its betweenness centrality once integrated into $G$, defined

as:

$$C_B(y) = \sum_{s \neq y \neq t \in V} \frac{\sigma_{st}(y)}{\sigma_{st}},$$

where $\sigma_{st}$ is the total number of shortest paths between nodes $s$ and $t$, and $\sigma_{st}(y)$ is the number of such paths passing through $y$. High $C_B(y)$ indicates that the idea bridges otherwise disconnected communities, a property correlated with transformative scientific discoveries (Ke et al., 2023). Such metrics provide a scalable, domain-agnostic way to estimate potential influence before real deployment, though they cannot capture context-specific or practical impact.

Beyond proxies, *execution-based* approaches directly implement generated hypotheses, mirroring scientific experimentation. These methods differ from feasibility checks in scope: feasibility asks "can this be done?", while impact asks "does it matter?"

The *AI Co-Scientist* (Gottweis et al., 2025) explicitly executes biomedical hypotheses to quantify real impact through end-to-end wet-lab experimentation. It autonomously proposes and tests hypotheses in three increasingly complex domains—drug repurposing, novel target discovery, and microbial evolution. The system systematically measures downstream biological outcomes—such as inhibition efficacy or regenerative response—to assess the practical significance of its discoveries. This rigorous execution-driven evaluation exemplifies how impact in scientific ideation can be assessed through direct experimentation rather than proxy metrics.

Beyond laboratory science, behavioral execution studies such as *MindShift* (Wu et al., 2024) and *LLM Counter-Speech* (Podolak et al., 2024) deploy AI-generated interventions in natural environments—digital health and social media—to quantify real behavioral change as a domain-specific key performance indicator (KPI) measuring practical impact. Similarly, large-scale agent-based frameworks like *AgentSociety* (Piao et al., 2025) model how new social or policy ideas propagate through simulated societies, tracking collective outcomes as impact proxies. In summary, impact assessment through execution-driven tests, behavioral outcome studies or graph-based scientometric, ensure that models are not merely producing ideas that are novel, diverse, or feasible, but ones that are also meaningful and capable of advancing science or society in a measurable way. However, such approaches are inherently limited: many scientific ideas are not directly executable, domain simulators exist only for select fields, and measurable outcomes often capture only the practical success of an idea rather than its deeper creative or theoretical significance. Consequently, nuanced dimensions of creativity and long-term scientific value still rely on human expertise and intuition, which remain the gold standard for assessing scientific ideas.

Table 4 summarizes the main evaluation paradigms—from human judgment to computational and execution-based methods—and their roles in assessing novelty, diversity, feasibility and impact.

## 7.2 Human Expert Evaluation

Because computational metrics cannot fully capture the impact, value, or nuanced creativity of scientific ideas, human experts remain the gold standard for evaluation in most scientific ideation studies. Expert reviewers can judge conceptual depth, originality, and methodological soundness—qualities that automated metrics still struggle to approximate.

Typical setups follow a *panel review* format, modeled after peer review. For example, *AIdeation* (Wang et al., 2025d) engaged professional concept designers to rate generated ideas

using **Likert scales** (e.g., 1–5 ratings) for creativity, feasibility, and satisfaction, supplemented by qualitative interviews. Similar setups are found in *Scideator* (Smith & Huang, 2025) and *Nova* (Hu et al., 2024), where domain specialists score hypotheses for novelty, plausibility, and usefulness, providing both quantitative and qualitative human feedback. More recently, large-scale human studies have emerged to improve robustness and generalizability of human evaluation. Notably, Si et al. (2024) evaluates LLM-generated research ideas using assessments from over 100 domain experts, explicitly separating judgments of novelty and usefulness.

To improve inter-rater reliability, several studies adopt the **Consensual Assessment Technique (CAT)** (Amabile, 1982), aggregating multiple expert scores into consensus ratings that reduce bias and align evaluation with established creativity research. At the most rigorous level, *AI Scientist v2* (Yamada et al., 2025) elevates human evaluation to formal peer review by submitting autonomously generated papers to top-tier conferences, where acceptance serves as the strongest validation of impact and scientific merit.

While indispensable, human evaluation is time-consuming, costly, and difficult to scale, with subjective variation among evaluators. This motivates the development of automated evaluators that can approximate expert intuition—capturing creativity, impact, and value at scale. The next section explores these efforts under the paradigm of *LLM-as-a-judge*.

### 7.3 LLM-as-a-Judge Evaluation

The *LLM-as-a-Judge* paradigm reframes large language models from generators to evaluators capable of approximating aspects of human evaluative judgment. Early benchmarks such as *MT-Bench and Chatbot Arena* (Zheng et al., 2023) showed that LLMs can reliably proxy human preferences and judgments even in zero-shot settings, establishing their use as scalable evaluators across domains. In scientific discovery, LLM-judges are now applied to assess plausibility, novelty, and impact of hypotheses, occupying a practical middle ground between purely computational metrics (which often miss conceptual depth), execution-based validation (which can be costly or infeasible), and labor-intensive human review. However, these systems do not replicate domain expertise or epistemic reasoning and should be interpreted as preference-aligned evaluators rather than substitutes for expert judgment.

As shown in Table 5, recent LLM-as-a-Judge systems can be broadly grouped into four methodological categories. The first involves **Prompt-based or inference-time judges**, where general-purpose LLMs are prompted or tool-called to evaluate idea quality along specific dimensions. The second category, **Fine-tuned judges**, trains LLMs on scientific peer-review corpora to better approximate expert judgment. The third, **Reinforcement-optimized judges**, refines evaluation behavior using reward models or preference optimization to enhance consistency and reasoning depth. Finally, **Hybrid frameworks** integrate training-time alignment with inference-time reasoning strategies, combining the model's internalized intuition with up-to-date external knowledge to produce more stable, nuanced, and human-aligned evaluations.

### 7.3.1 Prompt-based / Inference-time judges

LLMs' zero- and few-shot in-context reasoning enables them to act as first-line evaluators with minimal supervision. In the zero-shot setting, models are guided by reviewing rubrics encoded directly in prompts. For example, the GPT-4o-based *AI Scientist* (Lu et al., 2024a; Yamada et al., 2025) ap-

Table 5: Comparison of LLM-as-a-Judge paradigms for evaluating scientific ideas, showing their mechanisms, representative works, and key advantages and drawbacks.

| Judge category | Core idea | Example works | Strengths | Limitations |
|---|---|---|---|---|
| **Inference-time Judges (Prompt-based)** | Leverages structured prompts, retrieval engines, or multi-agent critique to approximate expert review without fine-tuning. | **AI Scientist** (Lu et al., 2024a), **LiveIdeaBench** (Ruan et al., 2024), **Chain-of-Idea** (Li et al., 2024a) | Low-cost, interpretable, and reproducible; easy to generalize across domains and tasks. | Highly sensitive to prompt phrasing and context; lacks consistent calibration across runs. |
| **Supervised Fine-tuned Judges** | Trains models on annotated peer-review corpora to internalize expert-like judgment and scoring patterns. | **CycleReviewer** (Weng et al., 2025a), **OpenReviewer** (Idahl & Ahmadi, 2025) | High correlation with expert judgments; improved consistency and interpretability. | Memorization instead of reasoning; static preferences; dependent on dataset coverage and quality; may inherit bias. |
| **RL and Reward-Tuned Judges** | Optimizes evaluative behavior via reinforcement learning or preference alignment for reasoning stability. | **HARPA** (Vasu et al., 2025), **ReviewRL** (Zeng et al., 2025), **REMOR** (Taechoyotin & Acuna, 2025) | Generalization and adaptivity; high correlation with expert judgments; encourages coherent, stable, and self-consistent evaluations. | Expensive to train; vulnerable to reward misspecification and preference drift. |
| **Hybrid (Training + Inference Integration)** | Combines fine-tuning with inference-time reasoning and external verifiers or factuality tools. | **DeepReview** (Zhu et al., 2025b), **HARPA** (Vasu et al., 2025) | Merges the generalization of training-based methods with the factual grounding of inference scaling methods. | Complex pipeline; requires maintaining multi-stage logic and external dependencies. |

plies NeurIPS-style reviewing criteria to generate structured assessments of generated manuscripts. *LiveIdeaBench* (Ruan et al., 2024) employs a multi-agent jury of LLM evaluators—each assigned distinct system prompts and roles—to rate generated ideas across originality, feasibility, fluency, flexibility, and significance. *ResearchAgent* (Baek et al., 2024) coordinates multiple reviewer agents to iteratively critique and refine hypotheses. Similarly, *Chain-of-Idea* (Li et al., 2024a) introduces a comparative "idea arena," evaluating pairs of hypotheses against each other rather than assigning absolute scores to reduce judgment drift and improve calibration.

Beyond pure prompting, a large body of work augments these evaluators with retrieval pipelines that ground LLM judgments in empirical evidence. Such *retrieval-based inference* enhances reliability in metrics like novelty, feasibility, and impact by contextualizing model reasoning with recent literature. For instance, the *Idea Novelty Checker* (Shahid et al., 2025) retrieves semantically related papers via dense embeddings, filters and re-ranks them with LLM reasoning across facets such as purpose and mechanism, achieving over 13% higher agreement with expert judgments. *Deep-Review* (Zhu et al., 2025b) extends this approach with multi-step retrieval and synthesis stages, generating structured "novelty verification" reports that summarize evidence gaps and methodological contributions.

Retrieval grounding has similarly advanced *feasibility assessment*. *Matter-of-Fact* (Jansen et al., 2025) benchmarks literature-grounded reasoning in materials science, testing whether claims are feasible given pre-publication data. *SciPIP* (Wang et al., 2024c) integrates citation-graph reasoning and semantic retrieval to estimate methodological plausibility and resource constraints, while *IdeaSynth* (Pu et al., 2025) decomposes ideas into problem–method–evaluation triplets and measures proximity to historically successful research, estimating both feasibility and experimental cost.

Finally, works like *aiXiv* (Zhang et al., 2025b) demonstrate large-scale retrieval-augmented evaluation pipelines for impact prediction, combining literature grounding with structured LLM judgment.

Overall, prompt-based and retrieval-augmented judges offer fast, reproducible, and low-cost evaluation with improved factual grounding. However, their reliability remains sensitive to prompt design, retrieval coverage, and base model reasoning quality.

### 7.3.2 Supervised fine-tuned judges

A more robust direction uses *SFT on large review datasets*, aligning LLMs to human scoring behavior. Here, models are exposed to richly annotated review datasets containing both scores and rationales, enabling them to internalize reviewing patterns. *CycleReviewer* (Weng et al., 2025a), trained as a generative reward model on the Review-5K dataset, achieves a decision accuracy of 74.24%, complementing human expertise with consistent automated scoring. Building on this, *OpenReviewer* Idahl & Ahmadi (2025) applies full-parameter SFT to a Llama-3.1-8B-Instruct model using the 80K reviews from top conferences, producing structured reviews with strong reasoning quality.

*DeepReview* (Zhu et al., 2025b) pushes SFT-based evaluation further through a significantly larger and more structured training corpus—*DeepReview-13K*—which integrates full research papers, multi-stage reasoning traces, and final human-aligned assessments. During inference, it employs retrieval-augmented review synthesis, self-verification, and reflection modules that enable iterative refinement of review reasoning. The resulting 14B-parameter model, despite its moderate size,

surpasses larger models like CycleReviewer-70B, GPT-o1 and Deepseek-R1 by dynamically scaling inference complexity, demonstrating that dataset design and structured supervision can outweigh pure model scale.

These SFT pipelines emphasize the importance of dataset curation and training objectives: the models learn scoring intuitions from reviewer' annotations, which yields high correlation with human judgments. However, recent studies highlight several limitations of SFT in capturing scientific novelty. (Shuieh et al., 2025) show that SFT often induces overreliance on spurious cues in the data, reducing robustness when those features fail.

### 7.3.3 Reinforcement-optimized judges

The third regime advances beyond SFT *memorization* toward *generalization* via RL (Chu et al., 2025). While SFT-based reviewers mimic human scoring behavior, they often overfit to annotation bias or linguistic patterns, limiting their ability to assess novel or out-of-distribution ideas. RL optimization instead trains judges to reason through evaluation criteria, promoting deeper alignment, calibration, and consistency.

Recent frameworks such as *J1* (Whitehouse et al., 2025) show that tailored RL objectives can internalize reasoning depth rather than surface agreement. J1 fine-tunes reviewers using reward signals from rubric adherence and multi-step self-consistency, outperforming SFT and DPO baselines in cross-domain generalization and inter-rater reliability. Similarly, *ReviewRL* (Zeng et al., 2025) warm-starts from SFT and applies Reinforce++ optimization for review quality and stability, achieving superior coherence, factuality, and fairness across venues like NeurIPS and ICLR.

*REMOR* (Taechoyotin & Acuna, 2025) extends this direction with multi-objective RL that balances factual accuracy, reasoning depth, and creativity encouragement. By coupling review text with reasoning traces and composite rewards—covering alignment, novelty, and consistency—it improves citation prediction accuracy and reduces verbosity compared to SFT-only baselines.

Finally, *HARPA* (Vasu et al., 2025) adapts the reinforcement-optimized paradigm to hypothesis generation. Its learned reward model scores ideas by testability and groundedness using prior experimental outcomes as feedback. HARPA-generated hypotheses outperform strong AI-researcher baselines in feasibility (+0.78) and groundedness (+0.85) on human ratings, achieving higher real-world execution success in the *CodeScientist* pipeline (20 vs. 11 successful runs). This illustrates how RL-trained evaluators can not only assess but also guide ideation toward scientifically meaningful discoveries. By directly optimizing for rating stability and agreement with human gold standards, RL-trained critics achieve more reliable judgments across domains.

In summary, RL introduces adaptive feedback loops that explicitly reward desirable evaluation behaviors such as coherence, fairness, and consistency. Studies such as J1, ReviewRL, and REMOR demonstrate that RL-trained judges produce more stable, reasoning-aligned, and domain-general evaluations—offering a more robust foundation for scientific idea assessment. However, even these post-training approaches have intrinsic limits when evaluations depend on external scientific evidence or knowledge-intensive reasoning.

### 7.3.4 Hybrid Frameworks

The most effective designs often combine multiple strategies. Metrics such as novelty, feasibility, or impact often require grounding in up-to-date literature, experiment outcomes, or citation graphs—information beyond what is encoded in model parameters. As a result, SFT or pure RL post-trained reviewers may exhibit confident but uninformed judgments, especially in fast-evolving or specialized research areas.

To overcome these constraints, hybrid evaluators integrate learned judgment policies with test-time reasoning and external tool use. *DeepReviewer* (Zhu et al., 2025b) exemplifies the fusion of training-based alignment and inference-time adaptability. It combines the structured evaluative intuition learned during supervised fine-tuning with external retrieval and self-verification mechanisms activated at inference, allowing the model to ground its reasoning dynamically in relevant literature. Through its scalable inference modes (Fast, Standard, Best), DeepReviewer-14B adjusts deliberation depth to match computational budgets, maintaining both efficiency and analytical rigor. This synergy between training priors and adaptive test-time reasoning enables DeepReviewer to outperform much larger models such as CycleReviewer-70B, producing richer and more accurate reviews with fewer tokens and strong resilience to adversarial perturbations—despite the absence of explicit robustness training. *ReviewRL* (Zeng et al., 2025) similarly augments its RL-trained critic with retrieval-augmented context (e.g., ArXivMCP) to ground evaluations in verifiable scientific evidence. Such hybrid systems balance the efficiency of automation with the rigor of human-like review, highlighting a consensus that future evaluation frameworks will need to fuse training-based alignment with external grounding to achieve both reliability and transparency.

Recent work comparing human and LLM-based reviewers highlights complementary strengths but clear gaps. Zero-shot systems such as Claude 3.5 Sonnet produce structured, comprehensive reviews yet lack the contextual depth and domain reasoning of humans (Renata & Lee, 2025). Similarly, GPT-4o performs well in summarizing contributions but misses weaknesses and critical insights (Li et al., 2025b). Broader evaluations show that while GPT-based models reach over 60% accuracy in review prediction, they remain limited in long-context and analytical reasoning (Zhou et al., 2024). Other analyses further warn that LLMs, though reducing reviewer fatigue, risk amplifying bias and diminishing accountability(Hosseini & Horbach, 2023). Overall, LLM-based judges provide scalable and consistent first-pass evaluations but still require human calibration to ensure fairness, nuance, and contextual rigor (Table 5).

### 7.4 Challenges in Evaluation

Among the four Ps of creativity, the *Product* dimension is uniquely challenging: it represents the measurable outcome of creativity, whereas *Person*, *Process*, and *Press* serve as sources that shape how such products emerge (Kozbelt et al., 2010; Gruszka & Tang, 2017). For scientific idea generation, this means that evaluation is not just an auxiliary step but the central bottleneck—where questions of novelty, feasibility, diversity, and impact must ultimately be resolved. Yet current practices reveal deep limitations that hinder progress.

First, evaluation suffers from *subjectivity and inconsistency*. Human raters often diverge in their judgments depending on expertise, background, or expectations, and are prone to systematic biases—for instance, underrating AI-generated ideas or devaluing content once labeled as machine-produced (Glikson & Woolley, 2020; Langham et al., 2022; Zhu et al., 2025c). Automated proxies

Table 6: Scientific idea generation benchmarks and how they operationalize creativity dimensions.

| Benchmark | Task / Input | Evaluation protocol | Key strengths / limitations | Creativity dimensions |
|---|---|---|---|---|
| **Research-Bench** Liu et al. | • Inspiration retrieval
• Idea / hypothesis composition
• Ranking stage | • Stage-wise evaluation
• Ranking accuracy
• Grounding
• Human / LLM judges | + End-to-end discovery modeling
+ Explicit grounding
- Implicit scores | Novelty, Grounding |
| **IdeaBench** (Guo et al., 2024) | • Research ideas from paper context
• Title / abstract / references | • Expert or LLM judges
• Likert / pairwise ranking | + Explicit ideation focus
+ Scalable templates
- Qualitative feasibility
- Judge sensitivity | Novelty, Feasibility |
| **LiveIdea-Bench** (Ruan et al., 2024) | • Minimal context prompts
• Keyword / short input
• Divergent ideation test | • Panel LLM judges
• Multi-axis scoring
• Score aggregation | + Low-context creativity probe
+ Highly scalable
- Judge bias / drift
- Weak feasibility grounding | Novelty, Diversity, Surprise |
| **HypoBench** (Liu et al., 2025c) | • Data-driven hypotheses
• Real + synthetic tasks
• Known structure | • Ground-truth recovery
• Discovery rate metrics
• Human / LLM judges | + Strong validity signal
+ Quantitative evaluation
- Statistical focus
- Limited ideation | Validity, Testability |
| **MatDesign** (Kumbhar et al., 2025) | • Materials science domain
• Knowledge-constrained prompts
• Domain realism | • Expert-aligned criteria
• Feasibility, plausibility
• Impact, testability | + Rich feasibility checks
+ Realistic hypotheses
- Narrow domain
- High expert cost | Novelty, Feasibility, Testability |
| **Research-CodeBench** (Hua et al., 2025b) | • Code-level realization
• Implement recent methods
• Execution-based tasks | • Compile / run tests
• Functional correctness
• Quality checks | + Objective feasibility signal
+ Execution grounded
- No novelty measure
- Realization $\neq$ ideation | Feasibility, Correctness |
| **LLM-SRBench** 2025 | • Equation discovery
• Synthetic transformations
• Contam.-aware design | • Exact correctness
• Fit accuracy
• Robustness tests | + Formal validity
+ Strong testability
- Narrow discovery form
- Limited ideation scope | Validity, Testability |
| **Matter-ofFact** Jansen et al. | • Literature-supported claims
• Materials science domain | • Claim feasibility verification
• Consistency checks
• Human / LLM judges | + Explicit feasibility grounding
- Not generative by itself
- Domain-specific | Feasibility, Validity |

such as embedding similarity or LLM critics are not immune either, as they may replicate biases encoded in training data or drift with prompting (Laskar et al., 2024; Wang et al., 2025e).

A second challenge is the *lack of standardization*: most studies define their own evaluation metrics, leading to fragmented practices and results that are not reproducible or comparable across benchmarks (Gehrmann et al., 2022; Laskar et al., 2024). Table 6 illustrates this heterogeneity: benchmarks vary not only in task framing (idea vs. hypothesis generation vs. discovery vs. code realization), but also in evaluation protocols (expert panels, LLM-as-judge, ground-truth recovery, execution-based correctness) and in how common criteria like "novelty" or "feasibility" are operationalized. Moreover, most existing benchmarks evaluate creativity primarily at the level of the final artifact. Since the generative process is treated as a black box, these benchmarks cannot directly assess the source or mechanism of creativity, and instead focus on observable outcomes such as novelty, plausibility, or executability. As a consequence, many evaluations rely on short-term or proxy-based metrics, leaving the dynamics of idea generation and longer-term scientific impact largely unexamined. (Ismayilzada et al., 2025b). Fourth, there is the risk of *bias amplification*. Both human reviewers and LLM-based judges can propagate historical inequities, undervaluing unconventional or marginalized research directions unless carefully calibrated (Gehrmann et al., 2022). Finally, a persistent *scalability gap* remains unresolved: while expert review provides nuance and reliability, it is expensive and prone to fatigue effects, whereas automated metrics scale efficiently but fail to capture the depth of human judgment (Thelwall, 2025).

Together, these challenges highlight evaluation as the most urgent bottleneck for LLM-driven scientific discovery. Overcoming them will require hybrid strategies that combine transparent and standardized metrics, scalable automated proxies, and carefully calibrated human oversight. Only by strengthening the evaluation of the *Product* dimension can we ensure that generative models contribute not just more ideas, but ideas that are genuinely valuable for advancing science.

## 8 Ethical Concerns

While the integration of LLMs into scientific ideation substantially lowers the time and cost of hypothesis generation, experimental design, and exploratory research, it simultaneously introduces a range of ethical risks. As idea generation becomes automated and scalable, evaluation and verification must also be automated, since manual review cannot keep pace with the growing volume of outputs. This pressure is evident in contemporary peer review: NeurIPS 2025 received over 21,500 submissions, more than double its 2022 count (Foundation, 2025), far exceeding the capacity of available expert reviewers and motivating the use of automated evaluation tools such as LLM-as-a-judge (Section 7.3). However, as discussed in the previous section, reliable evaluation is fundamentally harder than generation. Evaluation must verify correctness, feasibility, evidential grounding, and must distinguish genuine creativity—combining novelty and scientific value—from fabrication, hallucination, or superficially convincing but flawed ideas. This difficulty gives rise to the risk of *plausible pseudo-science*, where LLM-generated ideas appear coherent and innovative yet rest on incorrect assumptions or unsound reasoning that automated evaluators struggle to detect (Li et al., 2025a). In addition, LLM-based ideation and evaluation systems may *amplify existing biases* in the scientific literature(Zhu et al., 2025a), preferentially favoring dominant methods, paradigms, or problem formulations while neglecting underexplored or unconventional directions, thereby reinforcing prevailing scientific dogmas. Beyond these modeling limitations, automation introduces *security risks*: recent work has demonstrated that LLM-based reviewers are vulnerable to prompt injection attacks (Collu et al., 2025; Lin, 2025; Zhu et al., 2025a; Theocharopoulos et al., 2025), in which adversarial text embedded in manuscripts manipulates automated reviews, highlighting how evaluation pipelines can be systematically exploited. More broadly, the use of LLMs for scientific ideation raises *dual-use* concerns, particularly in domains such as chemistry and biology, where generated ideas or procedures could be misused if insufficient safeguards are in place(Urbina et al., 2022). While recent studies and guidelines have begun to outline responsible practices for using AI in scientific research(Resnik & Hosseini, 2024; Knöchel et al., 2025), substantial challenges remain for governance and policymaking to adequately address verification robustness, bias, adversarial manipulation, and dual-use risks in automated scientific discovery.

# 9 Future Works and Research Gaps

## 9.1 Embracing the Open-Ended Nature of Scientific Discovery

Unlike bounded reasoning tasks such as mathematics, scientific discovery is intrinsically open-ended: the space of possible hypotheses is unbounded, goals evolve as new evidence arises, and reframing questions is often as important as solving them. Yet most current LLM-based methods—whether retrieval-augmented generation, prompt-driven creativity, or inference-time scaling—implicitly assume fixed objectives and closed task spaces. This mismatch suggests that the field has so far underappreciated the open-ended nature of scientific ideation, treating it as a well-posed reasoning benchmark rather than a continually expanding search.

Concepts from open-ended learning offer useful inspiration for future directions. Novelty search demonstrated that progress can emerge by prioritizing behavioral novelty over explicit objectives (Lehman & Stanley, 2011). Quality-diversity methods such as MAP-Elites show how exploration can be structured to discover diverse, high-performing solutions across a wide range of niches (Mouret & Clune, 2015). Co-evolutionary frameworks such as *POET* (Wang et al., 2019) and hard-exploration strategies like *Go-Explore* (Ecoffet et al., 2019; 2021) illustrate how agents can sustain long-term innovation by generating new tasks alongside new solutions, or by systematically revisiting and extending prior stepping stones. At larger scales, environments like *XLand* demonstrate that procedurally generated, open-ended multi-task worlds can produce generally capable agents through continual adaptation (Team et al., 2021). Earlier theoretical work such as *POW-ERPLAY* (Schmidhuber, 2011) also emphasized continual self-invented tasks as a foundation for unbounded discovery.

Adapting these ideas to LLM-based pipelines could mark a step change for scientific ideation. Instead of optimizing within static benchmarks, future systems should integrate mechanisms for continuous exploration, novelty seeking, and dynamic reframing, enabling them to generate not just candidate answers but entirely new scientific questions.

## 9.2 Establishing Standardized Benchmarks and Robust Evaluation Frameworks

Evaluation remains the most critical bottleneck for LLM-based scientific ideation. At present, nearly every study introduces its own pipeline for assessing novelty, diversity, or feasibility, which makes systematic comparison across approaches nearly impossible (Ismayilzada et al., 2025b; Gupta et al., 2024; Ruan et al., 2024). In terms of creativity theory, this corresponds to the *Product* dimension: creative outcomes are the dependent variables by which novelty, usefulness, and correctness are judged (Kozbelt et al., 2010). Yet current methods lack standardized and transparent metrics for these criteria, leaving evaluations subjective and inconsistent. Automated metrics such as embedding-based novelty scores (Wang et al., 2025e), diversity indices (Friedman & Dieng, 2023), or literature-grounded overlap detection (Shahid et al., 2025) are valuable but insufficient on their own. More importantly, human evaluation practices remain opaque: many papers fail to clearly report annotator expertise, evaluation protocols, or inter-rater reliability, leaving results subjective and difficult to reproduce (Gehrmann et al., 2022; Laskar et al., 2024).

A crucial step forward is the development of shared benchmarks and transparent evaluation pipelines. These should span multiple domains (e.g., biology, chemistry, mathematics, social sciences) and multiple levels of abstraction, from hypothesis plausibility to experimental feasibility.

Human evaluation in particular needs greater rigor: standardized reporting of evaluator backgrounds, annotation guidelines, and agreement metrics should become community norms. Beyond laboratory-style annotation, peer-review–style evaluations hold particular promise. For instance, the AI Scientist project submitted an AI-generated paper to a workshop as a form of external validation (Lu et al., 2024a), demonstrating how conference-style review could provide structured and credible human feedback. Similar efforts—whether via workshops, structured review panels, or crowd-sourced expert collectives—could provide scalable, transparent, and standardized human-in-the-loop evaluation.

Beyond human assessment, future work must also address limitations of *LLM-based automated evaluators*. Supervised fine-tuned judges can suffer from memorization and overfitting, limiting their ability to generalize to genuinely novel ideas. Recent directions suggest augmenting evaluators with *online retrieval* and *test-time reasoning*, allowing judgments to be grounded in up-to-date literature and refined through iterative evaluation rather than static pattern matching(Zhu et al., 2025b). More broadly, no single evaluator is sufficient for all criteria: novelty assessment benefits from retrieval against current research, feasibility is best evaluated via domain-specific executors or simulators, and overall correctness or coherence may be more reliably judged by fine-tuned expert models. This motivates a *modular, criterion-specific evaluation paradigm* rather than monolithic scoring.

Finally, an important open challenge is *spurious learning* in trained evaluators, where models may rely on superficial cues (e.g., topic popularity or stylistic novelty) instead of core scientific qualities. Addressing this requires techniques inspired by the spurious correlation mitigation literature, such as balanced or counterfactual sampling, re-weighting of evaluation data, and explicit control over correlated attributes during evaluator training(Yang et al., 2023). Incorporating these strategies is essential for building reliable and fair automated reviewers for scientific ideation.

By establishing community-accepted benchmarks, rigorous human evaluation practices, and robust automated evaluators, the field can move beyond isolated proof-of-concept systems toward results that are comparable, reproducible, and trusted by the broader scientific community.

### 9.3 Building Domain-Rich Simulators for Feasible and Scalable Training

Direct interaction with real scientific environments—such as wet labs, clinical trials, or field experiments—is often costly, slow, and in some cases unsafe. This poses a major barrier to RL and closed-loop discovery pipelines, which require frequent and structured feedback signals to improve. Developing high-fidelity, domain-rich simulators could therefore transform LLM-based scientific ideation by providing safe, rapid, and scalable environments for hypothesis testing.

The role of simulators in accelerating progress is well established in other areas of AI. In robotics and control, physics-based simulators such as *MuJoCo* (Todorov et al., 2012) or *Isaac Gym* (Makoviychuk et al., 2021) have enabled breakthroughs in RL by allowing agents to practice millions of interactions before real-world deployment. In scientific discovery, similar trends are emerging: AtomAgents integrates alloy design with materials simulators for iterative hypothesis refinement (Ghafarollahi & Buehler, 2024), while *Robin* uses in silico biological assays and robotic wet-labs to close the loop between hypothesis generation and experimental validation (Ghareeb et al., 2025). More abstractly, multi-agent simulation environments such as *Casevo* (Jiang et al., 2024) show how

agent societies can serve as dynamic testbeds for evaluating ideation in complex social or political domains.

By building domain-specific simulators for materials, biology, physics, and socio-technical systems, the community can enable structured rewards, scalable pretraining, and safe exploration. These environments bridge symbolic reasoning and empirical science, accelerating the development of RL-based and hybrid discovery systems. Ultimately, simulators provide scaffolding for LLMs to generate, test, and refine hypotheses at scale—strengthening the *Person* and *Press* dimensions of creativity by offering feedback-rich environments that enhance agents' intrinsic scientific abilities.

### 9.4 Beyond Next-Token Prediction: Architectural Bottlenecks for Creativity

A final challenge concerns the architectural limits of current LLMs, which are fundamentally trained on the next-token prediction objective. Although this paradigm produces fluent language and strong surface-level reasoning, it inherently biases models toward local coherence and incremental extrapolation, discouraging bold conceptual leaps that scientific discovery often requires (Franceschelli & Musolesi, 2024; Bender et al., 2021; Marcus, 2022). This tendency toward "safe" continuations can stifle creativity and prevent models from venturing into genuinely novel scientific hypotheses. From the perspective of the 4Ps framework, this reflects a limitation of the *Person* dimension: the intrinsic capacities of the idea-generating system are constrained by its training objectives and inductive biases. Emerging research points to several promising alternatives Multi-token prediction frameworks, such as *Roll the Dice and Look Before You Leap* (Nagarajan et al., 2025), seek to reduce the local myopia of auto-regressive decoding by predicting multiple tokens jointly, improving global coherence. State-space sequence models such as *Mamba* demonstrate long-context stability and efficient memory handling, offering potential for sustained scientific reasoning (Gu & Dao, 2024; Halloran et al., 2024). Meanwhile, *Diffusion-based language models* (Nie et al., 2025; Li & Liang, 2022) depart entirely from the auto-regressive paradigm by framing text generation as iterative denoising, which can encourage more flexible and globally optimized exploration. Hybrid approaches that combine these paradigms with transformers may provide the best path forward, balancing local fluency with global creativity.

Future research should therefore go beyond next-token prediction to explore whether such architectures—or their integration into hybrid systems—can unlock richer creative capacities and better reflect the open-ended, exploratory nature of scientific inquiry.

## 10    Conclusion

Large language models are beginning to reshape the earliest stages of the scientific discovery pipeline by serving as powerful engines for idea generation. Building on advances in reasoning paradigms such as inference-time scaling, reinforcement learning post-training, and multi-agent collaboration, researchers have shown that LLMs can retrieve, combine, and refine scientific concepts in ways that accelerate hypothesis formation. At the same time, frameworks of creativity highlight both the promise and the limitations of current systems: while combinatorial and exploratory creativity are increasingly within reach, truly transformational discovery remains elusive.

Our survey organized this rapidly evolving field into complementary methodological dimensions: **Knowledge augmentation**, which grounds ideation in external evidence; **Prompt-driven creativity**, which steers models toward novelty via careful role and constraint design; **Inference-time search**, which enables iterative refinement, branching exploration, and ensemble aggregation; **Collaborative multi-agent systems**, which simulate the dynamics of research teams; and **Parameter-level adaptations**, which fine-tune models for domain-specific reasoning and alignment. We also reviewed evaluation practices, showing how novelty, diversity, feasibility, and impact are currently measured through a patchwork of automated metrics and ad hoc human studies, with reproducibility and comparability remaining serious challenges.

Finally, viewed through the lens of creativity frameworks, several gaps become clear. Current methods concentrate on the *Process* and *Press* dimensions—structuring search and leveraging external knowledge—yet the *Person* and *Product* dimensions remain underdeveloped. Enhancing the person perspective may require rethinking training objectives, architectures, or moving from idea-level to agent-level search, as seen in open-ended paradigms. On the product side, the lack of standardized benchmarks and rigorous multi-metric evaluations leaves creativity assessment vague and inconsistent. Addressing these blind spots—by strengthening intrinsic generative capacities and grounding evaluation in transparent metrics—could move LLMs beyond proof-of-concept ideation toward genuine scientific contributions, evolving from research assistants into collaborators capable of pushing the frontiers of human knowledge.

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
