# OpenReview forum: "Large Language Models for Scientific Idea Generation: A Creativity-Centered Survey"
_TMLR — Accepted by TMLR_

### Review · Reviewer_1hm4 · 2025-11-30

**Summary Of Contributions:**

This paper presents a comprehensive survey of Large Language Models (LLMs) applied to the specific domain of Scientific Idea Generation. The authors propose a taxonomy that bridges machine learning methodologies with cognitive science frameworks, specifically organizing existing works through the lens of Rhodes’ 4Ps (Person, Process, Press, Product) and Boden’s three levels of creativity (Combinatorial, Exploratory, Transformational).

The survey categorizes current techniques into five distinct families: Knowledge Augmentation (Press), Prompt-based Steering (Press), Inference-time Search (Process), Multi-agent Collaboration (Process), and Parameter-level Adaptation (Person). Furthermore, it provides a detailed analysis of evaluation paradigms, distinguishing between computational metrics, execution-based validation, and human/LLM-based judgment.

The survey goes beyond reporting SOTA results to identify fundamental limitations. It powerfully argues that current methods are largely stuck at the "Combinatorial" or "Exploratory" levels of creativity. It identifies a critical research gap: that true "Transformational" creativity may be structurally limited by the next-token prediction paradigm itself , suggesting a necessary pivot toward open-ended learning and non-autoregressive architectures.

**Audience:**

Yes

**Audience Explanation:**

On the positive side, the paper provides a detailed discussion of Test-time Search, Reinforcement Learning (RL) Post-training, and Multi-Agent Debate. These are currently central technical pathways in the ML community for enhancing reasoning capabilities. By concretizing these general ML techniques within the specific scenario of scientific discovery, the survey offers valuable inspiration for researchers focused on algorithmic development.

However, the submission dedicates significant space to cognitive science theories, specifically Boden’s taxonomy and Rhodes’ 4Ps. Technical ML researchers might find this framing unnecessary or distracting.

**Broader Impact Concerns:**

While the paper discusses "Safety" and "Bias" briefly in the evaluation section, critical ethical implications of generative scientific ideation warrant a more explicit and dedicated Broader Impact Statement.

There is a significant risk that LLMs optimized for "novelty" and "plausibility"  might generate highly convincing but scientifically flawed hypotheses. If produced at scale, this could flood the scientific community with noise, wasting experimental resources and eroding trust in the peer-review system.

The paper highlights methods for generating chemicals, proteins, and experimental protocols (e.g., AtomAgents, CodeScientist). The potential for these systems to lower the barrier for designing novel pathogens, toxins, or hazardous materials is a severe safety concern that requires robust guardrails beyond standard alignment training.

As noted in this paper, RAG and SFT rely on existing literature. There is a concern that these systems will reinforce existing scientific dogmas, marginalize non-English or non-mainstream research, and create feedback loops where AI cites only highly-cited papers, stifling truly transformational diversity.

**Claims And Evidence:**

Yes

**Claims Explanation:**

The authors abstracts complex technical methodologies into high-level concepts, supporting these abstractions with clear, concrete examples. For instance, Scenario 5 effectively illustrates the abstract mechanisms of search strategies, clearly contrasting Local Search, Beam Search, and Tree Search in the specific task of "hallucination mitigation".

Some critiques due to lack of empirical grounding:

- The categorization of methods into combinatorial, exploratory, and transformational creativity feels idealistic. There is little evidence provided on "how to measure" the transition between these levels or how to empirically distinguish them in current LLM outputs.

- The mapping of technical methods to Rhodes' 4Ps framework (Person, Process, Press, Product) is interesting, but inherently interpretative and subjective. The "accuracy" of this taxonomy is argumentative rather than factual.

For example, the authors categorize inference-time scaling (search) under Process. However, it could be argued that since this involves self-correction and internal knowledge, it belongs to the Person dimension (internal capability). Similarly, knowledge augmentation is mapped to Press (environment), yet once data is retrieved, it functionally becomes part of the processing context. The mapping is not presented in a systematic or reproducible manner (e.g., a definitive matrix).

- The evidence supporting the superiority of complex methods is weak due to the state of the field. While the paper claims to outline directions toward "reliable" and "transformative" applications, the evaluation section explicitly admits to a "lack of standardized metrics" and "fragmented practices".

**Requested Changes:**

The current evaluation section admits to a "lack of standardized metrics" and mentions several benchmarks sporadically. To substantiate the claims regarding creativity and the 4Ps, I request the addition of a systematic summary table or matrix that organizes these key benchmarks (including other necessary benchmarks, e.g., ResearchBench, IdeaBench).

Expand the discussion on ethical risks specific to scientific discovery. This should explicitly cover "plausible pseudo-science" (convincing but flawed ideas), dual-use risks (especially in chemical/biological generation), and bias amplification where models might reinforce existing scientific dogmas.

Missing reference: Can LLMs Generate Novel Research Ideas? A Large-Scale Human Study with 100+ NLP Researchers. ICLR 2025.

---

> ### Author Response · Authors · 2026-01-24
>
> We thank the reviewer for their helpful and constructive feedback. Below, we address each point and describe the corresponding revisions made to the manuscript.
>
> ---
>
> ### **1. Systematic Benchmark Summary and Evaluation Landscape**
>
> In response, we add a **systematic benchmark summary table (Table 6)** that organizes representative evaluation benchmarks for scientific idea and hypothesis generation, including ResearchBench, IdeaBench, and related benchmarks. The table contrasts benchmarks along task formulation, evaluation protocol, strengths and limitations, and the creativity dimensions they operationalize.
>
> This structured comparison substantiates our discussion of the lack of standardization by showing that benchmarks differ substantially in how criteria such as novelty, feasibility, and impact are defined and measured (e.g., human judgment, LLM-as-judge, execution-based verification, or ground-truth recovery). The table further clarifies that existing benchmarks primarily evaluate creative *products*, while other dimensions of the 4Ps—particularly *process* and *person*—remain implicit or unobserved. We explicitly reference this table in the evaluation challenges section to ground claims about fragmented practices and proxy-based evaluation.
>
> ---
>
> ### **2. Expanded Discussion of Ethical Risks in Scientific Discovery**
>
> We expand the ethics section (Section 8) to explicitly address ethical risks specific to scientific discovery. This includes discussion of **plausible pseudo-science**, where LLM-generated ideas may appear coherent yet rely on flawed assumptions that automated evaluators fail to detect; **dual-use risks**, particularly in chemical and biological domains where generated hypotheses or procedures could be misused; and **bias amplification**, where LLMs may reinforce dominant paradigms present in training data, potentially marginalizing under-represented approaches.
>
> These additions clarify that ethical risks in LLM-based scientific ideation extend beyond general AI concerns and are closely tied to evaluation, verification, and governance challenges unique to scientific workflows.
>
> ---
>
> ### **3. Added Missing Reference on Human Evaluation of Novel Research Ideas**
>
> We add the missing work to the related work and human evaluation sections and cite it as an important large-scale empirical study on **human-assessed novelty and usefulness** of LLM-generated research ideas. This strengthens the empirical grounding of the human evaluation discussion.

---

### Review · Reviewer_kzrQ · 2025-12-25

**Summary Of Contributions:**

Contributions: This survey applies cognitive science frameworks to understand LLM-based scientific discovery. Using Boden's creativity taxonomy and Rhodes' 4Ps model, the authors organize existing methods into five families and uncover an important pattern. Current research has made strong progress on two dimensions: Process through inference-time scaling and multi-agent systems, and Press through knowledge augmentation and prompting. But two other dimensions remain underexplored. The Person dimension, which concerns how to build intrinsically creative models, and the Product dimension, which involves rigorously evaluating creative outputs, have received far less attention. Consequently, today's systems achieve combinatorial and sometimes exploratory creativity, but transformational creativity remains out of reach. The paper also critiques current evaluation practices, examining computational metrics, human judgment, and LLM-as-a-judge approaches while documenting persistent problems with subjectivity, standardization, and measuring long-term impact.

Strengths: The paper is very clear and well-organized. It brings in theoretical frameworks from cognitive science and notices that current work overemphasizes Process and Press while neglecting Person and Product, something that only becomes visible through the 4Ps lens. It uses Boden's creativity taxonomy together with Rhodes' 4Ps model for analysis and covers multiple scientific domains.

Weaknesses: The classification methodology lacks clarity. The paper does not clearly explain how to determine which creativity level a particular method belongs to, nor does it demonstrate how existing literature was systematically analyzed to support these classifications. Moreover, the boundary between claims supported by existing evidence and those based on theoretical speculation remains ambiguous.

**Audience:**

Yes

**Audience Explanation:**

This paper addresses LLM-based scientific discovery, connecting core ML methods to fundamental questions about creativity and generative capabilities. The framework and evaluation insights transfer broadly to any AI application balancing novelty with reliability.

**Broader Impact Concerns:**

No specific ethical concerns identified.

**Claims And Evidence:**

Yes

**Claims Explanation:**

This survey provides a clear theoretical framework for understanding the field by systematically aligning current LLM techniques with the 4Ps model and Boden's creativity taxonomy from cognitive science. The paper thoroughly examines the technical mechanisms underlying five major method families and uses Scenario Boxes to illustrate how different abstraction levels might operate in scientific research tasks. Additionally, the paper offers rigorous mathematical definitions for creativity evaluation metrics such as Relative Neighbor Density and the Vendi Score, and draws on extensive recent research from 2024-2025 to support its observation that current systems still face significant challenges in achieving transformational creativity. The paper's characterization of the current state of the field is well-documented and logically coherent.

**Requested Changes:**

1. The authors should explicitly clarify how methods are categorized within Boden’s taxonomy . The paper needs to explain the criteria used to judge whether a method belongs to combinatorial, exploratory, or transformational creativity, demonstrate how existing literature was systematically analyzed to support these classifications , and consistently distinguish between evidence-based observations and theoretical speculations throughout the text .
2. While the paper effectively identifies various problems in current evaluation practices, it would be significantly more valuable if the authors could propose constructive research directions to address these issues.

---

> ### Author Response · Authors · 2026-01-24
>
> We thank the reviewer for their helpful and constructive feedback. Below, we address each point in turn and describe the corresponding revisions made to the manuscript.
>
> ---
>
> ### **1. Clarifying Categorization Under Boden’s Taxonomy**
>
> In the revised manuscript, we expand the discussion of Boden’s taxonomy in the introduction (paragraph 5) and make the categorization criteria explicit. We clarify that methods are categorized based on the **mechanism by which they expand or reshape the ideation space**:
>
> - **Combinatorial creativity** corresponds to recombination of existing concepts without altering the underlying conceptual space.
> - **Exploratory creativity** reflects systematic search within a fixed but enriched space.
> - **Transformational creativity** involves modifying the structure of the space itself, often through interaction, critique, or learning.
>
> To ground these assignments in prior work, we add a **summary table (Table 3)** mapping each method category to (i) its expected creativity level under Boden’s taxonomy and (ii) its dominant creativity source under the 4Ps framework. The table consolidates evidence discussed in the text and distinguishes empirically supported trends from theoretically motivated hypotheses.
>
> Throughout the revision, we also update the language to consistently distinguish **evidence-based observations** from **conceptual interpretations**, reinforcing that Boden’s taxonomy is used as an analytical lens rather than a guarantee of creative outcomes.
>
> ---
>
> ### **2. Expanding Constructive Research Directions for Evaluation**
>
> We clarify that the paper already proposes concrete research directions in the *Future Works and Research Gaps* (Section 9.2), and we expand this discussion in the revision. Beyond advocating standardized benchmarks and transparent human evaluation, we add **actionable methodological directions** for improving automated evaluation pipelines.
>
> Specifically, we argue that evaluation should be **criterion-specific rather than monolithic**:  novelty assessment benefits from online retrieval against up-to-date literature, feasibility is best evaluated using domain-specific executors or simulators, and overall correctness or coherence may be more reliably judged by fine-tuned expert models.
> We further discuss limitations of **supervised fine-tuned LLM judges**, including memorization and spurious learning, and highlight recent mitigation strategies such as online retrieval augmentation, test-time reasoning, balanced or counterfactual sampling, and re-weighting. These additions clarify actionable paths toward more robust, scalable, and trustworthy evaluation of LLM-based scientific ideation.

---

### Review · Reviewer_pN3c · 2025-12-26

**Summary Of Contributions:**

This paper provides a structured and comprehensive analysis of the capabilities and limitations of large language models (LLMs) in scientific ideation, with a particular focus on the dimensions of valueness and novelty. A key contribution of the work is the explicit grounding of scientific ideation in cognitive science theories of creativity, notably Rhodes’ 4Ps framework (1961) and Boden’s taxonomy of creativity (2004), which offer a conceptual lens for categorizing and interpreting LLM-generated ideas.

Based on these frameworks, the paper organizes existing approaches to LLM-based scientific ideation into five categories: (1) knowledge and retrieval augmentation, (2) prompt-based distributional steering, (3) search and sampling expansions, (4) multi-agent and deliberative systems, and (5) parameter adaptation and learning. This taxonomy provides a clear and useful map of the current methodological landscape. In addition, the paper surveys existing metrics and evaluation methodologies for assessing generated scientific ideas, highlighting their strengths and limitations.

Overall, the paper is well-structured, clearly written, and covers a broad range of relevant literature. Its main strengths lie in conceptual clarity, systematic organization, and breadth of coverage. Potential weaknesses include the reliance on relatively classical cognitive creativity frameworks, some ambiguity in the distinction between novelty and creativity, and limited empirical validation for several high-level claims.

**Additional Comments:**

Overall, this is a well-written and timely paper that offers a valuable conceptual synthesis of LLM-based scientific ideation. With clearer definitions, stronger empirical grounding, and tighter integration between evaluation and method taxonomy, the paper has the potential to become a useful reference point for future work in this area.

**Audience:**

Yes

**Audience Explanation:**

The topic of LLM-based scientific ideation is highly relevant to the TMLR audience, particularly researchers interested in large language models, scientific discovery, creativity, and evaluation methodologies. The paper’s synthesis of cognitive creativity theories with modern LLM techniques is likely to be of interest to readers seeking conceptual frameworks for understanding and positioning recent work in this rapidly evolving area.

Moreover, the proposed taxonomy of methods and the discussion of evaluation practices provide a useful reference for both researchers developing new ideation systems and reviewers or practitioners assessing their outputs.

**Broader Impact Concerns:**

The paper is primarily a conceptual and methodological analysis and does not raise immediate ethical concerns. However, the authors may consider briefly discussing potential downstream risks of automated scientific ideation systems, such as the generation of misleading or low-quality hypotheses, over-reliance on model-generated ideas, or unequal access to advanced ideation tools. Addressing these points in a Broader Impact Statement would improve completeness but is not strictly required.

**Claims And Evidence:**

No

**Claims Explanation:**

While many of the paper’s descriptive and organizational claims are well supported by prior literature, several stronger claims are not fully substantiated by concrete empirical evidence within the paper. For example, the paper argues that overcoming the limitation in scenario 4 (Figure 4) in scientific ideation “requires inference-time scaling or multi-agent interaction, which actively expand the search space and allow for the emergence of more novel, cross-disciplinary ideas.” However, this claim is largely supported by citations and conceptual reasoning rather than direct experimental validation or systematic comparison across methods.

Similarly, while the paper discusses novelty and valueness as central dimensions of scientific ideation, it does not consistently demonstrate them through quantitative or comparative evaluation, fore example, how specific method classes improve one or both dimensions.

Overall, the evidence is sufficient for a high-level survey and conceptual analysis, but additional empirical grounding would be needed to convincingly support several of the paper’s broader claims.

**Requested Changes:**

1. Clarify the distinction between novelty and creativity (Critical).
The paper would benefit from a more precise conceptual separation between novelty and creativity. While some methods may demonstrably increase novelty, it is less clear whether and how they enhance creativity in a stronger cognitive sense. Explicit definitions and clearer operationalization would improve rigor.

2. Justify the choice of cognitive creativity frameworks (Non-critical).
The use of Rhodes’ 4Ps framework and Boden’s creativity taxonomy is conceptually helpful, but both frameworks are relatively classical. The authors could strengthen the paper by discussing why these frameworks remain appropriate today or by briefly acknowledging and situating more recent cognitive or computational creativity theories.

3. Provide clearer criteria for the five-category taxonomy (Critical).
The basis for categorizing methods into the five proposed classes should be made more explicit. In particular, it would be useful to explain how these categories are created and whether these categories fully cover the current literature on LLM-based scientific ideation.

4. Improve consistency between figure 3 and main text (Non-critical).
Figure 3 appears to present a four-level taxonomy, while the main text describes five categories. It would be helpful to clarify this discrepancy and ensure consistency between the figure and the text.

5. Strengthen empirical support for high-level claims (Critical).
Claims regarding the effectiveness of five classes of methods would benefit from additional empirical evidence, ablation studies, or at least a more systematic synthesis of existing experimental results.

6. Improve coherence between evaluation discussion and method taxonomy (Critical).
In Section 7, a direct comparison or evaluation of representative methods from each category using the discussed metrics would significantly strengthen the paper.

7. Explicitly link each method class to novelty and valueness (Non-critical).
A summary table mapping method categories to their expected or empirically observed impact on novelty and valueness would greatly enhance clarity and usability.

---

> ### Author Response · Authors · 2026-01-24
>
> We thank the reviewer for their thoughtful and constructive feedback. Below, we address each point and summarize the corresponding revisions.
>
> ---
>
> ### **1. Novelty vs. Creativity Distinction**
>
> We clarify that we adopt a **component-based view of creativity**, where creativity arises from the combination of novelty and valueness, following Boden and related computational creativity literature. In the revision, we correct instances where novelty and creativity were previously conflated and ensure that **novelty is treated as a necessary but not sufficient component of creativity**. We further clarify that increases in novelty alone do not imply improved creativity unless accompanied by gains in **valueness-related dimensions** such as correctness, feasibility, or utility (Section 1).
>
> ---
>
> ### **2. Justification for Using Boden and Rhodes Frameworks**
>
> As clarified in the introduction, we use cognitive creativity frameworks as *analytical lenses rather than mechanistic models*. Rhodes’ 4Ps framework and Boden’s taxonomy remain **high-level conceptual frameworks** with orthogonal dimensions that map naturally onto modern LLM-based systems.
>
> - **Rhodes’ 4Ps framework** provides a principled account of *where creativity resides* (Person, Process, Press, Product). This framework emerged empirically from Rhodes’ (1961) analysis of creativity definitions and has since been independently rediscovered across multiple research traditions (e.g., Stein, 1963; Mooney, 1963; MacKinnon, 1970; Odena & Welch, 2009), indicating a stable and recurring conceptual structure rather than a historical artifact (Jordanous, 2012).
>
> - **Boden’s taxonomy** distinguishes levels of creative output—combinatorial, exploratory, and transformational—well suited for **research-level scientific ideation**, unlike everyday-creativity frameworks such as 4C (Kaufman & Beghetto, 2009).
>
> We also note that **recent AI creativity studies** continue to employ these frameworks as domain-agnostic baselines (e.g., Nagarajan et al., 2025; Gu et al., 2025; Jackson et al., 2025; Ismayilzada et al., 2025b). In the revision, we briefly justify their continued relevance while acknowledging broader developments in computational creativity. Our intent is **not exclusivity**, but a **stable conceptual scaffold** for systematic comparison.
>
> ---
>
> ### **3. Criteria for the Five-Category Taxonomy**
>
> We clarify that the taxonomy is motivated by **how different LLM-based methods balance novelty with scientific soundness**. In the revised introduction, we explicitly state that the five categories are constructed using a **mechanism-based perspective**, grouping methods by how they intervene in the ideation process and how they are expected to affect novelty and valueness.
>
> We note that **closely related categorizations independently appear** in the LLM reasoning and agent literature (e.g., Huang et al., 2024; Wang et al., 2024; Zhang et al., 2024), supporting that our taxonomy reflects an **emerging shared structure** rather than an ad hoc partition. We also clarify that the taxonomy is **general and extensible**, not exhaustive.
>
> ---
>
> ### **4. Figure–Taxonomy Consistency**
>
> We correct an inconsistency by aligning **Figure 3** with the five-category taxonomy and updating the references accordingly.
>
> ---
>
> ### **5. Strengthening Empirical Support for High-Level Claims**
>
> As a survey, our goal is not to introduce new ablations but to *systematically synthesize existing empirical evidence*. We strengthen empirical grounding by:
>
> 1. Adding **representative empirical results** for each method category, discussed in the *Takeaway and Limitations* subsections (2.3, 3.5, 4.4, 5.3, 6.4).
> 2. Introducing **Table 3**, which consolidates representative works and **empirically observed trends**, explicitly linking categories to novelty and valueness.
> 3. Revising the text to **soften causal language** and consistently distinguish **empirical evidence from theoretical interpretation** (e.g., Section 3.5).
>
> ---
>
> ### **6. Coherence Between Evaluation and Method Taxonomy**
>
> We clarify that evaluation metrics are largely **method-agnostic**, as they assess the *product* of ideation. Nevertheless, in the revision we explicitly link each method category to **representative evaluation practices** used in prior work (e.g., human judgments, Elo scores, domain-specific rewards, peer-review–style evaluation), improving coherence while preserving metric generality. ( 2.3, 3.5, 4.4, 5.3, 6.4.)
>
> ---
>
> ### **7. Explicit Mapping to Novelty and Valueness**
>
> In the revision, we explicitly link each method class to its **expected and empirically observed effects on novelty and valueness**. **Table 3** maps the five categories to their primary creativity dimensions and representative empirical findings. We emphasize that these mappings reflect **observed trends rather than guarantees**, strengthening coherence between the taxonomy and evaluation discussion while maintaining the survey’s conceptual scope.

---

### Review · Reviewer_VBMt · 2026-01-05

**Summary Of Contributions:**

The paper presents a comprehensive survey of LLM based approaches for scientific idea generation. The problem of idea generation is framed as a dual objective problem, balancing novelty with scientific valueness.

To evaluate existing LLM based approaches, a methodical categorization is done as:
1. External knowledge augmentation
2. Prompt based distributional steering
3. Inference time scaling
4. Multi-agent collaboration
5. Parameter level adaptation
with each section evaluating sub-strategies and their limitations.

For interpretation, the authors leverage Boden's taxonomy and Rhodes 4P framework to organize the field. The paper also reviews current evaluation practices for scientific ideation, while outlining open challenges and future research directions.

Key strengths - the paper provides a clear, unifying framework to connect and evaluate diverse techniques for idea generation. It also highlights why evaluation remains difficult and why progress in benchmarks and validation frameworks is essential for the field.

A key area where the paper can provide additional value is that it often relies on plausible but not always empirically validated assumptions; incorporating more detailed empirical evidences will help readers better assess the claims.

**Audience:**

Yes

**Audience Explanation:**

Yes, researchers and practitioners working on LLM-based scientific discovery, automated research systems would find the paper's synthesis and framework useful.

**Broader Impact Concerns:**

Not applicable. The paper is a survey and does not introduce new models or deployable systems, and thus poses no major concerns.

**Claims And Evidence:**

Yes

**Claims Explanation:**

The survey is clear and generally careful in grounding its discussion in the literature. That being said, there are some higher-level interpretations - particularly around how certain techniques relate to deeper idea generation - that are based more on plausible reasoning than strong empirical evidence. Overall, the claims are credible, but they would benefit from clearer empirical support.

**Requested Changes:**

I find the paper suitable for acceptance in its current form. The points below are suggestions for improvement that could further strengthen the presentation and sharpen some of the arguments:
1. Section 2.3: The conclusion that knowledge augmentation primarily supports combinatorial creativity is reasonable but appears somewhat abrupt. Given the central role of Boden's taxonomy, it would help to make this mapping explicit earlier in the section by clearly indicating how semantic and relational retrieval align with creativity levels, rather than leaving the reader to infer it.
2. Section 3.3: The claim that structured prompting constitutes a step towards transformational creativity appears overstated under Boden's definition. Transformational creativity requires altering conceptual space, whereas the prompt based methods discussed primarily challenge assumptions within a fixed, pretrained representation. It would strengthen the paper to either qualify this claim more carefully or clarify the specific sense in which such prompting is considered transformational.
3. Section 4: The discussion suggested that tree-based search better preserves unconventional yet valuable hypotheses through backtracking. While plausible, this assumption is not directly supported by cited empirical studies. In practice, heuristic-guided tree search may still systematically deprioritize low-confidence or atypical ideas. Explicitly acknowledging this limitation would strengthen the analysis.
4. Section 7.1: While the surveyed metrics effectively operationalize proxies such as novelty, diversity, feasibility, and impact, it would be helpful to more explicitly acknowledge that these measures remain indirect proxies for creativity, rather than direct measurements of creative insight or conceptual innovation.
5. Section 7.3: The statement that LLM-as-a-Judge systems “approximate human reasoning” appears overstated. Existing benchmarks primarily demonstrate correlation with human preferences, not equivalence in reasoning or evaluative judgment-particularly in scientific contexts where domain intuition and epistemic uncertainty are central. Rephrasing this claim to avoid implying reasoning-level parity would improve rigor.

---

> ### Author Response · Authors · 2026-01-24
>
> We thank the reviewer for the thoughtful and constructive feedback. Below, we address each comment in turn.
>
> 1. **Knowledge Augmentation and Boden’s Taxonomy**
>    We agree that the mapping to Boden’s taxonomy should be made explicit earlier in the section. In the revision, we introduce an explicit grounding paragraph at the start of Section 2.3, clarifying that knowledge augmentation primarily supports *combinatorial creativity* by enabling recombination of existing concepts rather than expansion or transformation of the conceptual space. We further support this mapping with recent empirical evidence (e.g., Gu et al., 2025), which explicitly evaluates combinatorial creativity under retrieval-augmented generation.
>    In addition, we clarify how different retrieval strategies align with this creativity level: semantic retrieval facilitates recombination of closely related concepts, while relational or graph-based retrieval enables richer recombination across abstraction levels and domains. This makes the connection between retrieval mechanisms and creativity levels explicit throughout the section, rather than leaving it implicit at the conclusion.
>
> 2. **Structured Prompting and Transformational Creativity.**
>    We agree that, under Boden’s definition, strict transformational creativity requires alteration of the underlying conceptual space, which prompt-based methods do not achieve. In the revision, we soften our claim and clarify that structured prompting contributes only *procedurally* toward transformational behavior by reshaping effective constraints, assumptions, and search trajectories within a fixed pretrained space, rather than modifying the space itself (section 3.3). This distinction better aligns the discussion with Boden’s framework while preserving the empirical role of structured prompting in expanding exploratory behavior.
>
> 3. **Tree-Based Search and Preservation of Unconventional Hypotheses.**
>    We agree that the preservation of unconventional yet valuable hypotheses is not guaranteed by tree-based search alone and depends critically on heuristic signals and pruning strategies. In the revised manuscript, we clarify that tree-based methods offer a structural capacity for broader exploration—by maintaining multiple branches and enabling backtracking—rather than an inherent guarantee of retaining atypical ideas.
>    We further emphasize that the practical benefits of backtracking are conditional, and that tree-based search primarily expands the representational search space at increased computational cost. We add citations from planning and neural sequence generation literature documenting the irreversible pruning behavior of beam search and empirically demonstrating the role of backtracking in mitigating premature commitment (section 4.1.3 paragraph 3).
>
> 4. **Creativity Metrics as Proxies.**
>    We agree that metrics such as novelty, diversity, feasibility, and impact are indirect proxies for creativity rather than direct measurements of creative insight or conceptual innovation. In the revision, we make this distinction explicit by clarifying that computational and execution-based metrics operationalize observable properties of generated outputs and should be interpreted as comparative evaluation indicators, not as exhaustive or direct measures of creativity itself (section 7.1 paragraph 1).
>
> 5. **LLM-as-a-Judge Framing.**
>    We agree that describing LLM-as-a-Judge systems as “approximating human reasoning” was overstated. In the revised manuscript, we rephrase this claim to emphasize that such systems approximate human evaluative judgments and preferences rather than reasoning-level equivalence. We also explicitly note that existing benchmarks primarily demonstrate correlation with human preferences, and that LLM-based judges do not replace domain expertise or epistemic judgment, particularly in scientific contexts (section 7.3 paragraph 1).

---

### Decision · Action_Editor_TYmm · 2026-01-25

**Recommendation:** Accept as is

**Audience:**

Yes

**Audience Explanation:**

This work is highly relevant to the TMLR audience as it addresses the rapidly growing intersection of machine learning and automated scientific discovery. Researchers and practitioners focusing on reasoning, generative models, and agentic systems will find the integration of formal creativity frameworks with modern LLM techniques particularly valuable.

**Claims And Evidence:**

Yes

**Claims Explanation:**

The submission provides a comprehensive and structured survey of large language models for scientific idea generation that is well-grounded in the literature.